# Near-Optimal Reinforcement Learning with Self-Play

**Yu Bai**
Salesforce Research
yu.bai@salesforce.com

**Chi Jin**
Princeton University
chij@princeton.edu

**Tiancheng Yu**
MIT
yutc@mit.edu

## Abstract

This paper considers the problem of designing optimal algorithms for reinforcement learning in two-player zero-sum games. We focus on self-play algorithms which learn the optimal policy by playing against itself without any direct supervision. In a tabular episodic Markov game with $S$ states, $A$ max-player actions and $B$ min-player actions, the best existing algorithm for finding an approximate Nash equilibrium requires $\tilde{\mathcal{O}}(S^2AB)$ steps of game playing, when only highlighting the dependency on $(S, A, B)$. In contrast, the best existing lower bound scales as $\Omega(S(A + B))$ and has a significant gap from the upper bound. This paper closes this gap for the first time: we propose an optimistic variant of the *Nash Q-learning* algorithm with sample complexity $\tilde{\mathcal{O}}(SAB)$, and a new *Nash V-learning* algorithm with sample complexity $\tilde{\mathcal{O}}(S(A + B))$. The latter result matches the information-theoretic lower bound in all problem-dependent parameters except for a polynomial factor of the length of each episode. In addition, we present a computational hardness result for learning the best responses against a fixed opponent in Markov games—a learning objective different from finding the Nash equilibrium.

## 1   Introduction

A wide range of modern artificial intelligence challenges can be cast as a multi-agent reinforcement learning (multi-agent RL) problem, in which more than one agent performs sequential decision making in an interactive environment. Multi-agent RL has achieved significant recent success on traditionally challenging tasks, for example in the game of GO [30, 31], Poker [6], real-time strategy games [33, 22], decentralized controls or multiagent robotics systems [5], autonomous driving [27], as well as complex social scenarios such as hide-and-seek [3]. In many scenarios, the learning agents even outperform the best human experts .

Despite the great empirical success, a major bottleneck for many existing RL algorithms is that they require a tremendous number of samples. For example, the biggest AlphaGo Zero model is trained on tens of millions of games and took more than a month to train [31]. While requiring such amount of samples may be acceptable in simulatable environments such as GO, it is not so in other sample-expensive real world settings such as robotics and autonomous driving. It is thus important for us to understand the *sample complexity* in RL—how can we design algorithms that find a near optimal policy with a small number of samples, and what is the fundamental limit, i.e. the minimum number of samples required for any algorithm to find a good policy.

Theoretical understandings on the sample complexity for multi-agent RL are rather limited, especially when compared with single-agent settings. The standard model for a single-agent setting is an episodic Markov Decision Process (MDP) with $S$ states, and $A$ actions, and $H$ steps per episode. The best known algorithm can find an $\epsilon$ near-optimal policy in $\tilde{\Theta}(\text{poly}(H)SA/\epsilon^2)$ episodes, which matches the lower bound up to a single $H$ factor [1, 8]. In contrast, in multi-agent settings, the optimal sample complexity remains open even in the basic setting of two-player tabular Markov games

[28], where the agents are required to find the solutions of the games—the Nash equilibria. The best known algorithm, VI-ULCB, finds an $\epsilon$-approximate Nash equilibrium in $\tilde{\mathcal{O}}(\text{poly}(H)S^2AB/\epsilon^2)$ episodes [2], where $B$ is the number of actions for the other player. The information theoretical lower bound is $\Omega(\text{poly}(H)S(A+B)/\epsilon^2)$. Specifically, the number of episodes required for the algorithm scales quadratically in both $S$ and $(A, B)$, and exhibits a gap from the linear dependency in the lower bound. This motivates the following question:

**Can we design algorithms with near-optimal sample complexity for learning Markov games?**

In this paper, we present the first line of near-optimal algorithms for two-player Markov games that match the aforementioned lower bound up to a $\text{poly}(H)$ factor. This closes the open problem for achieving the optimal sample complexity in all $(S, A, B)$ dependency. Our algorithm learns by playing against itself without requiring any direct supervision, and is thus a *self-play* algorithm.

## 1.1 Our contributions

- We propose an optimistic variant of *Nash Q-learning* [11], and prove that it achieves sample complexity $\tilde{\mathcal{O}}(H^5SAB/\epsilon^2)$ for finding an $\epsilon$-approximate Nash equilibrium in two-player Markov games (Section 3). Our algorithm builds optimistic upper and lower estimates of $Q$-values, and computes the *Coarse Correlated Equilibrium* (CCE) over this pair of $Q$ estimates as its execution policies for both players.

- We design a new algorithm—*Nash V-learning*—for finding approximate Nash equilibria, and show that it achieves sample complexity $\tilde{\mathcal{O}}(H^6S(A+B)/\epsilon^2)$ (Section 4). This improves upon Nash Q-learning in case $\min\{A, B\} > H$. It is also the first result that matches the minimax lower bound up to only a $\text{poly}(H)$ factor. This algorithm builds optimistic upper and lower estimates of $V$-values, and features a novel combination of Follow-the-Regularized-Leader (FTRL) and standard Q-learning algorithm to determine its execution policies.

- Apart from finding Nash equilibria, we prove that learning the best responses of fixed opponents in Markov games is as hard as learning parity with noise—a notoriously difficult problem that is believed to be computationally hard (Section 5). As a corollary, this hardness result directly implies that achieving sublinear regret against *adversarial opponents* in Markov games is also computationally hard, a result that first appeared in [25]. This in turn rules out the possibility of designing efficient algorithms for finding Nash equilibria by running no-regret algorithms for each player separately.

In addition to above contributions, this paper also features a novel approach of extracting *certified policies*—from the estimates produced by reinforcement learning algorithms such as Nash Q-learning and Nash V-learning—that are *certified* to have similar performance as Nash equilibrium policies, even when facing against their best response (see Section 3 for more details). We believe this technique could be of broader interest to the community.

## 1.2 Related Work

**Markov games**  Markov games (or stochastic games) are proposed in the early 1950s [28]. They are widely used to model multi-agent RL. Learning the Nash equilibria of Markov games has been studied in classical work [18, 19, 11, 10], where the transition matrix and reward are assumed to be known, or in the asymptotic setting where the number of data goes to infinity. These results do not directly apply to the non-asymptotic setting where the transition and reward are unknown and only a limited amount of data are available for estimating them.

A recent line of work tackles self-play algorithms for Markov games in the non-asymptotic setting with strong reachability assumptions. Specifically, Wei et al. [35] assumes no matter what strategy one agent sticks to, the other agent can always reach all states by playing a certain policy, and Jia et al. [13], Sidford et al. [29] assume access to simulators (or generative models) that enable the agent to directly sample transition and reward information for any state-action pair. These settings ensure that all states can be reached directly, so no sophisticated exploration is not required.

Very recently, [2, 36] study learning Markov games without these reachability assumptions, where exploration becomes essential. However, both results suffer from highly suboptimal sample complexity. We compare them with our results in Table 1. The results of [36] also applies to the linear

Table 1: Sample complexity (the required number of episodes) for algorithms to find $\epsilon$-approximate Nash equilibrium policies in zero-sum Markov games.

| Algorithm | Sample Complexity | Runtime |
|---|---|---|
| VI-ULCB [2] | $\tilde{\mathcal{O}}(H^4 S^2 AB/\epsilon^2)$ | PPAD-complete |
| VI-explore [2] | $\tilde{\mathcal{O}}(H^5 S^2 AB/\epsilon^2)$ | Polynomial |
| OMVI-SM [36] | $\tilde{\mathcal{O}}(H^4 S^3 A^3 B^3/\epsilon^2)$ | |
| Optimistic Nash Q-learning | $\tilde{\mathcal{O}}(H^5 SAB/\epsilon^2)$ | |
| Optimistic Nash V-learning | $\tilde{\mathcal{O}}(H^6 S(A+B)/\epsilon^2)$ | |
| Lower Bound [14, 2] | $\Omega(H^3 S(A+B)/\epsilon^2)$ | - |

function approximation setting. We remark that the R-max algorithm [4] does provide provable guarantees for learning Markov game, even in the setting of playing against the adversarial opponent, but using a definition of regret that is weaker than the standard regret. Their result does not imply any sample complexity result for finding Nash equilibrium policies.

**Adversarial MDP**  Another line of related work focuses on provably efficient algorithms for *adversarial MDPs*. Most work in this line considers the setting with adversarial rewards [38, 26, 15], because adversarial MDP with changing dynamics is computationally hard even under full-information feedback [37]. These results do not direcly imply provable self-play algorithms in our setting, because the opponent in Markov games can affect both the reward and the transition.

**Single-agent RL**  There is a rich literature on reinforcement learning in MDPs [see e.g. 12, 24, 1, 7, 32, 14]. MDP is a special case of Markov games, where only a single agent interacts with a stochastic environment. For the tabular episodic setting with nonstationary dynamics and no simulators, the best sample complexity achieved by existing model-based and model-free algorithms are $\tilde{\mathcal{O}}(H^3 SA/\epsilon^2)$ [1] and $\tilde{\mathcal{O}}(H^4 SA/\epsilon^2)$ [14], respectively, where $S$ is the number of states, $A$ is the number of actions, $H$ is the length of each episode. Both of them (nearly) match the lower bound $\Omega(H^3 SA/\epsilon^2)$ [12, 23, 14].

## 2 Preliminaries

We consider zero-sum Markov Games (MG) [28, 18], which are also known as stochastic games in the literature. Zero-sum Markov games are generalization of standard Markov Decision Processes (MDP) into the two-player setting, in which the *max-player* seeks to maximize the total return and the *min-player* seeks to minimize the total return.

Formally, we denote a tabular episodic Markov game as $\text{MG}(H, \mathcal{S}, \mathcal{A}, \mathcal{B}, \mathbb{P}, r)$, where $H$ is the number of steps in each episode, $\mathcal{S}$ is the set of states with $|\mathcal{S}| \leq S$, $(\mathcal{A}, \mathcal{B})$ are the sets of actions of the max-player and the min-player respectively, $\mathbb{P} = \{\mathbb{P}_h\}_{h \in [H]}$ is a collection of transition matrices, so that $\mathbb{P}_h(\cdot|s, a, b)$ gives the distribution over states if action pair $(a, b)$ is taken for state $s$ at step $h$, and $r = \{r_h\}_{h \in [H]}$ is a collection of reward functions, and $r_h: \mathcal{S} \times \mathcal{A} \times \mathcal{B} \to [0, 1]$ is the deterministic reward function at step $h$. [1]

In each episode of this MG, we start with a *fixed initial state* $s_1$. Then, at each step $h \in [H]$, both players observe state $s_h \in \mathcal{S}$, and the max-player picks action $a_h \in \mathcal{A}$ while the min-player picks action $b_h \in \mathcal{B}$ simultaneously. Both players observe the actions of the opponents, receive reward $r_h(s_h, a_h, b_h)$, and then the environment transitions to the next state $s_{h+1} \sim \mathbb{P}_h(\cdot|s_h, a_h, b_h)$. The episode ends when $s_{H+1}$ is reached.

**Markov policy, value function** A *Markov* policy $\mu$ of the max-player is a collection of $H$ functions $\{\mu_h : \mathcal{S} \to \Delta_{\mathcal{A}}\}_{h \in [H]}$, which maps from a state to a distribution of actions. Here $\Delta_{\mathcal{A}}$ is the probability simplex over action set $\mathcal{A}$. Similarly, a policy $\nu$ of the min-player is a collection of $H$ functions $\{\nu_h : \mathcal{S} \to \Delta_{\mathcal{B}}\}_{h \in [H]}$. We use the notation $\mu_h(a|s)$ and $\nu_h(b|s)$ to present the probability of taking action $a$ or $b$ for state $s$ at step $h$ under Markov policy $\mu$ or $\nu$ respectively.

We use $V_h^{\mu,\nu} : \mathcal{S} \to \mathbb{R}$ to denote the value function at step $h$ under policy $\mu$ and $\nu$, so that $V_h^{\mu,\nu}(s)$ gives the expected cumulative rewards received under policy $\mu$ and $\nu$, starting from $s$ at step $h$:

$$V_h^{\mu,\nu}(s) := \mathbb{E}_{\mu,\nu}\left[\left. \sum_{h'=h}^{H} r_{h'}(s_{h'}, a_{h'}, b_{h'}) \right| s_h = s \right]. \tag{1}$$

We also define $Q_h^{\mu,\nu} : \mathcal{S} \times \mathcal{A} \times \mathcal{B} \to \mathbb{R}$ to denote $Q$-value function at step $h$ so that $Q_h^{\mu,\nu}(s,a,b)$ gives the cumulative rewards received under policy $\mu$ and $\nu$, starting from $(s,a,b)$ at step $h$:

$$Q_h^{\mu,\nu}(s,a,b) := \mathbb{E}_{\mu,\nu}\left[\left. \sum_{h'=h}^{H} r_{h'}(s_{h'}, a_{h'}, b_{h'}) \right| s_h = s, a_h = a, b_h = b \right]. \tag{2}$$

For simplicity, we use notation of operator $\mathbb{P}_h$ so that $[\mathbb{P}_h V](s,a,b) := \mathbb{E}_{s' \sim \mathbb{P}_h(\cdot|s,a,b)} V(s')$ for any value function $V$. We also use notation $[\mathbb{D}_\pi Q](s) := \mathbb{E}_{(a,b) \sim \pi(\cdot,\cdot|s)} Q(s,a,b)$ for any action-value function $Q$. By definition of value functions, we have the Bellman equation

$$Q_h^{\mu,\nu}(s,a,b) = (r_h + \mathbb{P}_h V_{h+1}^{\mu,\nu})(s,a,b), \qquad V_h^{\mu,\nu}(s) = (\mathbb{D}_{\mu_h \times \nu_h} Q_h^{\mu,\nu})(s)$$

for all $(s,a,b,h) \in \mathcal{S} \times \mathcal{A} \times \mathcal{B} \times [H]$. We define $V_{H+1}^{\mu,\nu}(s) = 0$ for all $s \in \mathcal{S}_{H+1}$.

**Best response and Nash equilibrium** For any Markov policy of the max-player $\mu$, there exists a *best response* of the min-player, which is a Markov policy $\nu^\dagger(\mu)$ satisfying $V_h^{\mu,\nu^\dagger(\mu)}(s) = \inf_\nu V_h^{\mu,\nu}(s)$ for any $(s,h) \in \mathcal{S} \times [H]$. Here the infimum is taken over all possible policies which are not necessarily Markovian (we will define later in this section). We define $V_h^{\mu,\dagger} := V_h^{\mu,\nu^\dagger(\mu)}$. By symmetry, we can also define $\mu^\dagger(\nu)$ and $V_h^{\dagger,\nu}$. It is further known (cf. [9]) that there exist Markov policies $\mu^\star, \nu^\star$ that are optimal against the best responses of the opponents, in the sense that

$$V_h^{\mu^\star,\dagger}(s) = \sup_\mu V_h^{\mu,\dagger}(s), \qquad V_h^{\dagger,\nu^\star}(s) = \inf_\nu V_h^{\dagger,\nu}(s), \qquad \text{for all } (s,h).$$

We call these optimal strategies $(\mu^\star, \nu^\star)$ the Nash equilibrium of the Markov game, which satisfies the following minimax equation: [2]

$$\sup_\mu \inf_\nu V_h^{\mu,\nu}(s) = V_h^{\mu^\star,\nu^\star}(s) = \inf_\nu \sup_\mu V_h^{\mu,\nu}(s).$$

Intuitively, a Nash equilibrium gives a solution in which no player has anything to gain by changing only her own policy. We further abbreviate the values of Nash equilibrium $V_h^{\mu^\star,\nu^\star}$ and $Q_h^{\mu^\star,\nu^\star}$ as $V_h^\star$ and $Q_h^\star$. We refer readers to Appendix A for Bellman optimality equations for values of best responses or Nash equilibria.

**General (non-Markovian) policy** In certain situations, it is beneficial to consider general, history-dependent policies that are not necessarily Markovian. A *(general) policy* $\mu$ of the max-player is a set of $H$ maps $\mu := \left\{\mu_h : \mathbb{R} \times (\mathcal{S} \times \mathcal{A} \times \mathcal{B} \times \mathbb{R})^{h-1} \times \mathcal{S} \to \Delta_{\mathcal{A}}\right\}_{h \in [H]}$, from a random number $z \in \mathbb{R}$ and a history of length $h$—say $(s_1, a_1, b_1, r_1, \cdots, s_h)$, to a distribution over actions in $\mathcal{A}$. By symmetry, we can also define the (general) policy $\nu$ of the min-player, by replacing the action set $\mathcal{A}$ in the definition by set $\mathcal{B}$. The random number $z$ is sampled from some underlying distribution $\mathcal{D}$, but may be shared among all steps $h \in [H]$.

For a pair of general policy $(\mu, \nu)$, we can still use the same definitions (1) to define their value $V_1^{\mu,\nu}(s_1)$ at step 1. We can also define the best response $\nu^\dagger(\mu)$ of a general policy $\mu$ as the minimizing policy so that $V_1^{\mu,\dagger}(s_1) \equiv V_1^{\mu,\nu^\dagger(\mu)}(s_1) = \inf_\nu V_h^{\mu,\nu}(s_1)$ at step 1. We remark that the best response of a general policy is not necessarily Markovian.

**Algorithm 1** Optimistic Nash Q-learning

---

1: **Initialize:** for any $(s, a, b, h)$, $\overline{Q}_h(s, a, b) \leftarrow H$, $\underline{Q}_h(s, a, b) \leftarrow 0$, $N_h(s, a, b) \leftarrow 0$,
$\qquad\qquad \pi_h(a, b|s) \leftarrow 1/(AB)$.
2: **for** episode $k = 1, \dots, K$ **do**
3: $\quad$ receive $s_1$.
4: $\quad$ **for** step $h = 1, \dots, H$ **do**
5: $\qquad$ take action $(a_h, b_h) \sim \pi_h(\cdot, \cdot|s_h)$.
6: $\qquad$ observe reward $r_h(s_h, a_h, b_h)$ and next state $s_{h+1}$.
7: $\qquad t = N_h(s_h, a_h, b_h) \leftarrow N_h(s_h, a_h, b_h) + 1$.
8: $\qquad \overline{Q}_h(s_h, a_h, b_h) \leftarrow (1 - \alpha_t)\overline{Q}_h(s_h, a_h, b_h) + \alpha_t(r_h(s_h, a_h, b_h) + \overline{V}_{h+1}(s_{h+1}) + \beta_t)$
9: $\qquad \underline{Q}_h(s_h, a_h, b_h) \leftarrow (1 - \alpha_t)\underline{Q}_h(s_h, a_h, b_h) + \alpha_t(r_h(s_h, a_h, b_h) + \underline{V}_{h+1}(s_{h+1}) - \beta_t)$
10: $\qquad \pi_h(\cdot, \cdot|s_h) \leftarrow \text{CCE}(\overline{Q}_h(s_h, \cdot, \cdot), \underline{Q}_h(s_h, \cdot, \cdot))$
11: $\qquad \overline{V}_h(s_h) \leftarrow (\mathbb{D}_{\pi_h}\overline{Q}_h)(s_h); \quad \underline{V}_h(s_h) \leftarrow (\mathbb{D}_{\pi_h}\underline{Q}_h)(s_h)$.

---

**Learning Objective**   There are two possible learning objectives in the setting of Markov games. The first one is to find the best response for a fixed opponent. Without loss of generality, we consider the case where the learning agent is the max-player, and the min-player is the opponent.

**Definition 1** ($\epsilon$-approximate best response)**.**  For an opponent with an fixed unknown general policy $\nu$, a general policy $\hat{\mu}$ is the $\epsilon$-**approximate best response** if $V_1^{\dagger,\nu}(s_1) - V_1^{\hat{\mu},\nu}(s_1) \le \epsilon$.

The second goal is to find a Nash equilibrium of the Markov games. We measure the suboptimality of any pair of general policies $(\hat{\mu}, \hat{\nu})$ using the gap between their performance and the performance of the optimal strategy (i.e. Nash equilibrium) when playing against the best responses respectively:

$$V_1^{\dagger,\hat{\nu}}(s_1) - V_1^{\hat{\mu},\dagger}(s_1) = \left[ V_1^{\dagger,\hat{\nu}}(s_1) - V_1^{\star}(s_1) \right] + \left[ V_1^{\star}(s_1) - V_1^{\hat{\mu},\dagger}(s_1) \right]$$

**Definition 2** ($\epsilon$-approximate Nash equilibrium)**.**  A pair of general policies $(\hat{\mu}, \hat{\nu})$ is an $\epsilon$-**approximate Nash equilibrium**, if $V_1^{\dagger,\hat{\nu}}(s_1) - V_1^{\hat{\mu},\dagger}(s_1) \le \epsilon$.

Loosely speaking, Nash equilibria can be viewed as "the best responses to the best responses". In most applications, they are the ultimate solutions to the games. In Section 3 and 4, we present sharp guarantees for learning an approximate Nash equilibrium with near-optimal sample complexity. However, rather surprisingly, learning a best response in the worst case is more challenging than learning the Nash equilibrium. In Section 5, we present a computational hardness result for learning an approximate best response.

## 3   Optimistic Nash Q-learning

In this section, we present our first algorithm *Optimistic Nash Q-learning* and its corresponding theoretical guarantees.

**Algorithm part I: learning values**   Our algorithm *Optimistic Nash Q-learning* (Algorithm 1) is an optimistic variant of Nash Q-learning [11]. For each step in each episode, it (a) takes actions according to the previously computed policy $\pi_h$, and observes the reward and next state, (b) performs incremental updates on Q-values, and (c) computes new greedy policies and updates $V$-values. Part (a) is straightforward; we now focus on explaining part (b) and part (c).

In part (b), the incremental updates on Q-values (Line 8, 9) are almost the same as standard Q-learning [34], except here we maintain two separate Q-values—$\overline{Q}_h$ and $\underline{Q}_h$, as upper and lower confidence versions respectively. We add and subtract a bonus term $\beta_t$ in the corresponding updates, which depends on $t = N_h(s_h, a_h, b_h)$—the number of times $(s_h, a_h, b_h)$ has been visited at step $h$. We pick parameter $\alpha_t$ and $\beta_t$ as follows for some large constant $c$, and log factors $\iota$:

$$\alpha_t = (H+1)/(H+t), \qquad \beta_t = c\sqrt{H^3 \iota / t} \qquad (3)$$

In part (c), our greedy policies are computed using a *Coarse Correlated Equilibrium* (CCE) subroutine, which is first introduced by [36] to solve Markov games using value iteration algorithms. For

---

**Algorithm 2** Certified Policy $\hat{\mu}$ of Nash Q-learning

---

1: sample $k \leftarrow \text{Uniform}([K])$.
2: **for** step $h = 1, \ldots, H$ **do**
3:     observe $s_h$, and take action $a_h \sim \mu_h^k(\cdot|s_h)$.
4:     observe $b_h$, and set $t \leftarrow N_h^k(s_h, a_h, b_h)$.
5:     sample $m \in [t]$ with $\mathbb{P}(m = i) = \alpha_t^i$.
6:     $k \leftarrow k_h^m(s_h, a_h, b_h)$

---

any pair of matrices $\overline{Q}, \underline{Q} \in [0, H]^{A \times B}$, $\text{CCE}(\overline{Q}, \underline{Q})$ returns a distribution $\pi \in \Delta_{\mathcal{A} \times \mathcal{B}}$ such that

$$\mathbb{E}_{(a,b) \sim \pi} \overline{Q}(a, b) \geq \max_{a^\star} \mathbb{E}_{(a,b) \sim \pi} \overline{Q}(a^\star, b) \tag{4}$$

$$\mathbb{E}_{(a,b) \sim \pi} \underline{Q}(a, b) \leq \min_{b^\star} \mathbb{E}_{(a,b) \sim \pi} \underline{Q}(a, b^\star)$$

It can be shown that a CCE always exists, and it can be computed by linear programming in polynomial time (see Appendix B for more details).

Now we are ready to state an intermediate guarantee for optimistic Nash Q-learning. We assume the algorithm has played the game for $K$ episodes, and we use $V^k, Q^k, N^k, \pi^k$ to denote values, visitation counts, and policies *at the beginning* of the $k$-th episode in Algorithm 1.

**Lemma 3.** *For any $p \in (0, 1]$, choose hyperparameters $\alpha_t, \beta_t$ as in (3) for a large absolute constant $c$ and $\iota = \log(SABT/p)$. Then, with probability at least $1 - p$, Algorithm 1 has following guarantees*

- $\overline{V}_h^k(s) \geq V_h^\star(s) \geq \underline{V}_h^k(s)$ *for all* $(s, h, k) \in \mathcal{S} \times [H] \times [K]$.

- $(1/K) \cdot \sum_{k=1}^K (\overline{V}_1^k - \underline{V}_1^k)(s_1) \leq \mathcal{O}\left(\sqrt{H^5 SAB\iota/K}\right)$.

Lemma 3 makes two statements. First, it claims that the $\overline{V}_h^k(s)$ and $\underline{V}_h^k(s)$ computed in Algorithm 1 are indeed upper and lower bounds of the value of the Nash equilibrium. Second, Lemma 3 claims that the averages of the upper bounds and the lower bounds are also very close to the value of Nash equilibrium $V_1^\star(s_1)$, where the gap decrease as $1/\sqrt{K}$. This implies that in order to learn the value $V_1^\star(s_1)$ up to $\epsilon$-accuracy, we only need $\mathcal{O}(H^5 SAB\iota/\epsilon^2)$ episodes.

However, Lemma 3 has a significant drawback: it only guarantees the learning of the *value* of Nash equilibrium. It does not imply that the policies $(\mu^k, \nu^k)$ used in Algorithm 1 are close to the Nash equilibrium, which requires the policies to have a near-optimal performance even against their best responses. This is a major difference between Markov games and standard MDPs, and is the reason why standard techniques from the MDP literature does not apply here. To resolve this problem, we propose a novel way to extract a certified policy from the optimistic Nash Q-learning algorithm.

**Algorithm part II: certified policies**    We describe our procedure of executing the certified policy $\hat{\mu}$ of the max-player is described in Algorithm 2. Above, $\mu_h^k, \nu_h^k$ denote the marginal distributions of $\pi_h^k$ produced in Algorithm 1 over action set $\mathcal{A}, \mathcal{B}$ respectively. We also introduce the following quantities that directly induced by $\alpha_t$:

$$\alpha_t^0 := \prod_{j=1}^t (1 - \alpha_j), \ \alpha_t^i := \alpha_i \prod_{j=i+1}^t (1 - \alpha_j) \tag{5}$$

whose properties are listed in the following Lemma 11. Especially, $\sum_{i=1}^t \alpha_t^i = 1$, so $\{\alpha_t^i\}_{i=1}^t$ defines a distribution over $[t]$. We use $k_h^m(s, a, b)$ to denote the index of the episode where $(s, a, b)$ is observed in step $h$ for the $m$-th time. The certified policy $\hat{\nu}$ of the min-player is easily defined by symmetry. We note that $\hat{\mu}, \hat{\nu}$ are clearly general policies, but they are no longer Markov policies.

The intuitive reason why such policy $\hat{\mu}$ defined in Algorithm 2 is certified by Nash Q-learning algorithm, is because the update equation in line 8 of Algorithm 1 and equation (5) gives relation:

$$\overline{Q}_h^k(s, a, b) = \alpha_t^0 H + \sum_{i=1}^t \alpha_t^i \left[ r_h(s, a, b) + \overline{V}_{h+1}^{k_h^i(s,a,b)}(s_{h+1}^{k_h^i(s,a,b)}) + \beta_i \right]$$

---

**Algorithm 3** Optimistic Nash V-learning (the max-player version)

---

1: **Initialize:** for any $(s, a, b, h)$, $\overline{V}_h(s) \leftarrow H$, $\overline{L}_h(s, a) \leftarrow 0$, $N_h(s) \leftarrow 0$, $\mu_h(a|s) \leftarrow 1/A$.
2: **for** episode $k = 1, \ldots, K$ **do**
3:     receive $s_1$.
4:     **for** step $h = 1, \ldots, H$ **do**
5:         take action $a_h \sim \mu_h(\cdot|s_h)$, observe the action $b_h$ from opponent.
6:         observe reward $r_h(s_h, a_h, b_h)$ and next state $s_{h+1}$.
7:         $t = N_h(s_h) \leftarrow N_h(s_h) + 1$.
8:         $\overline{V}_h(s_h) \leftarrow \min\{H, (1 - \alpha_t)\overline{V}_h(s_h) + \alpha_t(r_h(s_h, a_h, b_h) + \overline{V}_{h+1}(s_{h+1}) + \overline{\beta}_t)\}$.
9:         **for** all $a \in \mathcal{A}$ **do**
10:             $\overline{\ell}_h(s_h, a) \leftarrow [H - r_h(s_h, a_h, b_h) - \overline{V}_{h+1}(s_{h+1})]\mathbb{I}\{a_h = a\}/[\mu_h(a_h|s_h) + \overline{\eta}_t]$.
11:             $\overline{L}_h(s_h, a) \leftarrow (1 - \alpha_t)\overline{L}_h(s_h, a) + \alpha_t\overline{\ell}_h(s_h, a)$.
12:         set $\mu_h(\cdot|s_h) \propto \exp[-(\overline{\eta}_t/\alpha_t)\overline{L}_h(s_h, \cdot)]$.

---

This certifies the good performance against the best responses if the max-player plays a mixture of policies $\{\mu_{h+1}^{k_h^i(s,a,b)}\}_{i=1}^t$ at step $h + 1$ with mixing weights $\{\alpha_t^i\}_{i=1}^t$ (see Appendix C.2 for more details). A recursion of this argument leads to the certified policy $\hat{\mu}$—a nested mixture of policies.

We now present our main result for Nash Q-learning, using the certified policies $(\hat{\mu}, \hat{\nu})$.

**Theorem 4** (Sample Complexity of Nash Q-learning). *For any $p \in (0, 1]$, choose hyperparameters $\alpha_t, \beta_t$ as in (3) for large absolute constant $c$ and $\iota = \log(SABT/p)$. Then, with probability at least $1 - p$, if we run Nash Q-learning (Algorithm 1) for $K$ episodes where*

$$K \geq \Omega\left(H^5 SAB\iota/\epsilon^2\right),$$

*the certified policies $(\hat{\mu}, \hat{\nu})$ (Algorithm 2) will be $\epsilon$-approximate Nash, i.e. $V_1^{\dagger, \hat{\nu}}(s_1) - V_1^{\hat{\mu}, \dagger}(s_1) \leq \epsilon$.*

Theorem 4 asserts that if we run the optimistic Nash Q-learning algorithm for more than $\mathcal{O}(H^5 SAB\iota/\epsilon^2)$ episodes, the certified policies $(\hat{\mu}, \hat{\nu})$ extracted using Algorithm 2 will be $\epsilon$-approximate Nash equilibrium (Definition 2).

We make two remarks. First, the executions of the certified policies $\hat{\mu}, \hat{\nu}$ require the storage of $\{\mu_h^k\}$ and $\{\nu_h^k\}$ for all $k, h \in [H] \times [K]$. This makes the space complexity of our algorithm scales up linearly in the total number of episodes $K$. Second, Q-learning style algorithms (especially online updates) are crucial in our analysis for achieving sample complexity linear in $S$. They enjoy the property that every sample is only been used once, on the value function that is independent of this sample. In contrast, value iteration type algorithms do not enjoy such an independence property, which is why the best existing sample complexity scales as $S^2$ [2]. [3]

## 4 Optimistic Nash V-learning

In this section, we present our new algorithm *Optimistic Nash V-learning* and its corresponding theoretical guarantees. This algorithm improves over Nash Q-learning in sample complexity from $\tilde{\mathcal{O}}(SAB)$ to $\tilde{\mathcal{O}}(S(A + B))$, when only highlighting the dependency on $S, A, B$.

**Algorithm description**    Nash V-learning combines the idea of Follow-The-Regularized-Leader (FTRL) in the bandit literature with the Q-learning algorithm in reinforcement learning. This algorithm does not require extra information exchange between players other than standard game playing, thus can be ran separately by the two players. We describe the max-player version in Algorithm 3. See Algorithm 7 in Appendix D for the min-player version, where $\underline{V}_h$, $\underline{L}_h$, $\nu_h$, $\underline{\eta}_t$ and $\underline{\beta}_t$ are defined symmetrically.

For each step in each episode, the algorithm (a) first takes action according to $\mu_h$, observes the action of the opponent, the reward, and the next state, (b) performs an incremental update on $\overline{V}$, and (c)

**Algorithm 4** Certified Policy $\hat{\mu}$ of Nash V-learning

1: sample $k \leftarrow \text{Uniform}([K])$.
2: **for** step $h = 1, \ldots, H$ **do**
3:     observe $s_h$, and set $t \leftarrow N_h^k(s_h)$.
4:     sample $m \in [t]$ with $\mathbb{P}(m = i) = \alpha_t^i$.
5:     $k \leftarrow k_h^m(s_h)$.
6:     take action $a_h \sim \mu_h^k(\cdot|s_h)$.

---

updates policy $\mu_h$. The first two parts are very similar to Nash Q-learning. In the third part, the agent first computes $\overline{\ell}_h(s_h, \cdot)$ as the importance weighted estimator of the current loss. She then computes the weighted cumulative loss $\overline{L}_h(s_h, \cdot)$. Finally, the policy $\mu_h$ is updated using FTRL principle:

$$\mu_h(\cdot|s_h) \leftarrow \text{argmin}_{\mu \in \Delta_{\mathcal{A}}} \ \overline{\eta}_t \langle \overline{L}_h(s_h, \cdot), \mu \rangle + \alpha_t \text{KL}(\mu \| \mu_0)$$

Here $\mu_0$ is the uniform distribution over all actions $\mathcal{A}$. Solving above minimization problem gives the update equation as in Line 12 in Algorithm 3. In multi-arm bandit, FTRL can defend against adversarial losses, with regret independent of the number of the opponent's actions. This property turns out to be crucial for Nash V-learning to achieve sharper sample complexity than Nash Q-learning (see the analog of Lemma 3 in Lemma 15).

Similar to Nash Q-learning, we also propose a new algorithm (Algorithm 4) to extract a certified policy from the optimistic Nash V-learning algorithm. The certified policies are again non-Markovian. We choose all hyperparameters as follows, for some large constant $c$, and log factors $\iota$.

$$\alpha_t = \frac{H+1}{H+t}, \quad \overline{\eta}_t = \sqrt{\frac{\log A}{At}}, \quad \underline{\eta}_t = \sqrt{\frac{\log B}{Bt}}, \quad \overline{\beta}_t = c\sqrt{\frac{H^4 A \iota}{t}}, \quad \underline{\beta}_t = c\sqrt{\frac{H^4 B \iota}{t}}, \quad (6)$$

We now present our main result on the sample complexity of Nash V-learning.

**Theorem 5** (Sample Complexity of Nash V-learning). *For any $p \in (0, 1]$, choose hyperparameters as in (6) for large absolute constant $c$ and $\iota = \log(SABT/p)$. Then, with probability at least $1 - p$, if we run Nash V-learning (Algorithm 3 and 7) for $K$ episodes with*

$$K \geq \Omega\left(H^6 S(A+B)\iota/\epsilon^2\right),$$

*its induced policies $(\hat{\mu}, \hat{\nu})$ (Algorithm 4) will be $\epsilon$-approximate Nash, i.e. $V_1^{\dagger, \hat{\nu}}(s_1) - V_1^{\hat{\mu}, \dagger}(s_1) \leq \epsilon$.*

Theorem 4 claims that if we run the optimistic Nash V-learning for more than $\mathcal{O}(H^6 S(A+B)\iota/\epsilon^2)$ episodes, the certified policies $(\hat{\mu}, \hat{\nu})$ extracted from Algorithm 4 will be $\epsilon$-approximate Nash (Definition 2). Nash V-learning is the first algorithm of which the sample complexity matches the information theoretical lower bound $\Omega(H^3 S(A+B)/\epsilon^2)$ up to $\text{poly}(H)$ factors and logarithmic terms.

## 5   Hardness for Learning the Best Response

In this section, we present a computational hardness result for computing the best response against an opponent with a fixed unknown policy. We further show that this implies the computational hardness result for achieving sublinear regret in Markov games when playing against adversarial opponents, which rules out a popular approach to design algorithms for finding Nash equilibria.

We first remark that if the opponent is restricted to only play Markov policies, then learning the best response is as easy as learning a optimal policy in the standard single-agent Markov decision process, where efficient algorithms are known to exist. Nevertheless, when the opponent can as well play any policy which may be non-Markovian, we show that finding the best response against those policies is computationally challenging.

We say an algorithm is a *polynomial time algorithm for learning the best response* if for any policy of the opponent $\nu$, and for any $\epsilon > 0$, the algorithm finds the $\epsilon$-approximate best response of policy $\nu$ (Definition 1) with probability at least $1/2$, in time polynomial in $S, H, A, B, \epsilon^{-1}$.

We can show the following hardness result for finding the best response in polynomial time.

**Theorem 6** (Hardness for learning the best response). *There exists a Markov game with deterministic transitions and rewards defined for any horizon $H \geq 1$ with $S = 2$, $A = 2$, and $B = 2$, such that if there exists a polynomial time algorithm for learning the best response for this Markov game, then there exists a polynomial time algorithm for learning parity with noise (see problem description in Appendix E).*

We remark that learning parity with noise is a notoriously difficult problem that has been used to design efficient cryptographic schemes. It is conjectured by the community to be hard.

**Conjecture 7** ([16]). *There is no polynomial time algorithm for learning party with noise.*

Theorem 6 with Conjecture 7 demonstrates the fundamental difficulty—if not strict impossibility—of designing a polynomial time for learning the best responses in Markov games. The intuitive reason for such computational hardness is that, while the underlying system has Markov transitions, the opponent can play policies that encode long-term correlations with non-Markovian nature, such as parity with noise, which makes it very challenging to find the best response. It is known that learning many other sequential models with long-term correlations (such as hidden Markov models or partially observable MDPs) is as hard as learning parity with noise [20].

## 5.1 Hardness for Playing Against Adversarial Opponent

Theorem 6 directly implies the difficulty for achieving sublinear regret in Markov games when playing against adversarial opponents in Markov games. Our construction of hard instances in the proof of Theorem 6 further allows the adversarial opponent to only play Markov policies in each episode. Since playing against adversarial opponent is a different problem with independent interest, we present the full result here.

Without loss of generality, we still consider the setting where the algorithm can only control the max-player, while the min-player is an adversarial opponent. In the beginning of every episode $k$, both players pick their own policies $\mu^k$ and $\nu^k$, and execute them throughout the episode. The adversarial opponent can possibly pick her policy $\nu^k$ *adaptive* to all the observations in the earlier episodes.

We say an algorithm for the learner is a *polynomial time no-regret algorithm* if there exists a $\delta > 0$ such that for *any* adversarial opponent, and any fixed $K > 0$, the algorithm outputs policies $\{\mu^k\}_{k=1}^K$ which satisfies the following, with probability at least $1/2$, in time polynomial in $S, H, A, B, K$.

$$\text{Regret}(K) = \sup_\mu \sum_{k=1}^K V_1^{\mu,\nu^k}(s_1) - \sum_{k=1}^K V_1^{\mu^k,\nu^k}(s_1) \leq \text{poly}(S, H, A, B)K^{1-\delta} \tag{7}$$

Theorem 6 directly implies the following hardness result for achieving no-regret against adversarial opponents, a result that first appeared in [25].

**Corollary 8** (Hardness for playing against adversarial opponent). *There exists a Markov game with deterministic transitions and rewards defined for any horizon $H \geq 1$ with $S = 2$, $A = 2$, and $B = 2$, such that if there exists a polynomial time no-regret algorithm for this Markov game, then there exists a polynomial time algorithm for learning parity with noise (see problem description in Appendix E). The claim remains to hold even if we restrict the adversarial opponents in the Markov game to be non-adaptive, and to only play Markov policies in each episode.*

Similar to Theorem 6, Corollary 8 combined with Conjecture 7 demonstrates the fundamental difficulty of designing a polynomial time no-regret algorithm against adversarial opponents for Markov games.

**Implications on algorithm design for finding Nash Equilibria** Corollary 8 also rules out a natural approach for designing efficient algorithms for finding approximate Nash equilibrium through combining two no-regret algorithms. In fact, it is not hard to see that if the min-player also runs a non-regret algorithm, and obtain a regret bound symmetric to (7), then summing the two regret bounds shows the mixture policies $(\hat{\mu}, \hat{\nu})$—which assigns uniform mixing weights to policies $\{\mu^k\}_{k=1}^K$ and $\{\nu^k\}_{k=1}^K$ respectively—is an approximate Nash equilibrium. Corollary 8 with Conjecture 7 claims that any algorithm designed using this approach is not a polynomial time algorithm.

## Broader Impact

As this is a theoretical contribution, we do not envision that our direct results will have a tangible societal impact. Our broader line of inquiry could impact a line of thinking about how to design more sample-efficient algorithms for multi-agent reinforcement learning, which could be useful towards making artificial intelligence more resource and energy efficient.

## Acknowledgments

TY is partially supported by NSF BIGDATA grant IIS1741341.

## Footnotes

[1] We assume the rewards in $[0, 1]$ for normalization. Our results directly generalize to randomized reward functions, since learning the transition is more difficult than learning the reward.

[2]The minimax theorem here is different from the one for matrix games, i.e. $\max_\phi \min_\psi \phi^\top A \psi = \min_\psi \max_\phi \phi^\top A \psi$ for any matrix $A$, since here $V_h^{\mu,\nu}(s)$ is in general not bilinear in $\mu, \nu$.

[3]Despite [1] provides techniques to improve the sample complexity from $S^2$ to $S$ for value iteration in MDP, the same techniques can not be applied to Markov games due to the unique challenge that, in Markov games, we aim at finding policies that are good against their best responses.

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
