[Supplementary Material]

# A Bellman Equations for Markov Games

In this section, we present the Bellman equations for different types of values in Markov games.

**Fixed policies.** For any pair of Markov policy $(\mu, \nu)$, by definition of their values in (1) (2), we have the following Bellman equations:

$$Q_h^{\mu,\nu}(s,a,b) = (r_h + \mathbb{P}_h V_{h+1}^{\mu,\nu})(s,a,b), \qquad V_h^{\mu,\nu}(s) = (\mathbb{D}_{\mu_h \times \nu_h} Q_h^{\mu,\nu})(s)$$

for all $(s,a,b,h) \in \mathcal{S} \times \mathcal{A} \times \mathcal{B} \times [H]$, where $V_{H+1}^{\mu,\nu}(s) = 0$ for all $s \in \mathcal{S}_{H+1}$.

**Best responses.** For any Markov policy $\mu$ of the max-player, by definition, we have the following Bellman equations for values of its best response:

$$Q_h^{\mu,\dagger}(s,a,b) = (r_h + \mathbb{P}_h V_{h+1}^{\mu,\dagger})(s,a,b), \qquad V_h^{\mu,\dagger}(s) = \inf_{\nu \in \Delta_\mathcal{B}} (\mathbb{D}_{\mu_h \times \nu} Q_h^{\mu,\dagger})(s),$$

for all $(s,a,b,h) \in \mathcal{S} \times \mathcal{A} \times \mathcal{B} \times [H]$, where $V_{H+1}^{\mu,\dagger}(s) = 0$ for all $s \in \mathcal{S}_{H+1}$.

Similarly, for any Markov policy $\nu$ of the min-player, we also have the following symmetric version of Bellman equations for values of its best response:

$$Q_h^{\dagger,\nu}(s,a,b) = (r_h + \mathbb{P}_h V_{h+1}^{\dagger,\nu})(s,a,b), \qquad V_h^{\dagger,\nu}(s) = \sup_{\mu \in \Delta_\mathcal{A}} (\mathbb{D}_{\mu \times \nu_h} Q_h^{\dagger,\nu})(s).$$

for all $(s,a,b,h) \in \mathcal{S} \times \mathcal{A} \times \mathcal{B} \times [H]$, where $V_{H+1}^{\dagger,\nu}(s) = 0$ for all $s \in \mathcal{S}_{H+1}$.

**Nash equilibria.** Finally, by definition of Nash equilibria in Markov games, we have the following Bellman optimality equations:

$$Q_h^\star(s,a,b) = (r_h + \mathbb{P}_h V_{h+1}^\star)(s,a,b)$$
$$V_h^\star(s) = \sup_{\mu \in \Delta_\mathcal{A}} \inf_{\nu \in \Delta_\mathcal{B}} (\mathbb{D}_{\mu \times \nu} Q_h^\star)(s) = \inf_{\nu \in \Delta_\mathcal{B}} \sup_{\mu \in \Delta_\mathcal{A}} (\mathbb{D}_{\mu \times \nu} Q_h^\star)(s).$$

for all $(s,a,b,h) \in \mathcal{S} \times \mathcal{A} \times \mathcal{B} \times [H]$, where $V_{H+1}^\star(s) = 0$ for all $s \in \mathcal{S}_{H+1}$.

# B Properties of Coarse Correlated Equilibrium

Recall the definition for CCE in our main paper (4), we restate it here after rescaling. For any pair of matrix $P, Q \in [0,1]^{n \times m}$, the subroutine $\mathrm{CCE}(P,Q)$ returns a distribution $\pi \in \Delta_{n \times m}$ that satisfies:

$$\mathbb{E}_{(a,b) \sim \pi} P(a,b) \geq \max_{a^\star} \mathbb{E}_{(a,b) \sim \pi} P(a^\star, b) \tag{8}$$
$$\mathbb{E}_{(a,b) \sim \pi} Q(a,b) \leq \min_{b^\star} \mathbb{E}_{(a,b) \sim \pi} Q(a, b^\star)$$

We make three remarks on CCE. First, a CCE always exists since a Nash equilibrium for a general-sum game with payoff matrices $(P,Q)$ is also a CCE defined by $(P,Q)$, and a Nash equilibrium always exists. Second, a CCE can be efficiently computed, since above constraints (8) for CCE can be rewritten as $n + m$ linear constraints on $\pi \in \Delta_{n \times m}$, which can be efficiently resolved by standard linear programming algorithm. Third, a CCE in general-sum games needs not to be a Nash equilibrium. However, a CCE in zero-sum games is guaranteed to be a Nash equalibrium.

**Proposition 9.** *Let $\pi = \mathrm{CCE}(Q,Q)$, and $(\mu, \nu)$ be the marginal distribution over both players' actions induced by $\pi$. Then $(\mu, \nu)$ is a Nash equilibrium for payoff matrix $Q$.*

*Proof of Proposition 9.* Let $N^\star$ be the value of Nash equilibrium for $Q$. Since $\pi = \mathrm{CCE}(Q,Q)$, by definition, we have:

$$\mathbb{E}_{(a,b) \sim \pi} Q(a,b) \geq \max_{a^\star} \mathbb{E}_{(a,b) \sim \pi} Q(a^\star, b) = \max_{a^\star} \mathbb{E}_{b \sim \nu} Q(a^\star, b) \geq N^\star$$
$$\mathbb{E}_{(a,b) \sim \pi} Q(a,b) \leq \min_{b^\star} \mathbb{E}_{(a,b) \sim \pi} Q(a, b^\star) = \min_{b^\star} \mathbb{E}_{a \sim \mu} Q(a, b^\star) \leq N^\star$$

This gives:

$$\max_{a^\star} \mathbb{E}_{b \sim \nu} Q(a^\star, b) = \min_{b^\star} \mathbb{E}_{a \sim \mu} Q(a, b^\star) = N^\star$$

which finishes the proof. $\square$

Intuitively, a CCE procedure can be used in Nash Q-learning for finding an approximate Nash equilibrium, because the values of upper confidence and lower confidence—$\overline{Q}$ and $\underline{Q}$ will be eventually very close, so that the preconditions of Proposition 9 becomes approximately satisfied.

# C   Proof for Nash Q-learning

In this section, we present proofs for results in Section 3.

We denote $V^k, Q^k, \pi^k$ for values and policies *at the beginning* of the $k$-th episode. We also introduce the following short-hand notation $[\widehat{\mathbb{P}}_h^k V](s,a,b) := V(s_{h+1}^k)$.

We will use the following notations several times later: suppose $(s,a,b)$ was taken at the in episodes $k^1, k^2, \ldots$ at the $h$-th step. Since the definition of $k^i$ depends on the tuple $(s,a,b)$ and $h$, we will show the dependence explicitly by writing $k_h^i(s,a,b)$ when necessary and omit it when there is no confusion. We also define $N_h^k(s,a,b)$ to be the number of times $(s,a,b)$ has been taken *at the beginning* of the $k$-th episode. Finally we denote $n_h^k = N_h^k\left(s_h^k, a_h^k, b_h^k\right)$.

The following lemma is a simple consequence of the update rule in Algorithm 1, which will be used several times later.

**Lemma 10.** *Let* $t = N_h^k(s,a,b)$ *and suppose* $(s,a,b)$ *was previously taken at episodes* $k^1, \ldots, k^t < k$ *at the $h$-th step. The update rule in Algorithm 1 is equivalent to the following equations.*

$$\overline{Q}_h^k(s,a,b) = \alpha_t^0 H + \sum_{i=1}^t \alpha_t^i \left[ r_h(s,a,b) + \overline{V}_{h+1}^{k^i}(s_{h+1}^{k^i}) + \beta_i \right] \tag{9}$$

$$\underline{Q}_h^k(s,a,b) = \sum_{i=1}^t \alpha_t^i \left[ r_h(s,a,b) + \underline{V}_{h+1}^{k^i}(s_{h+1}^{k^i}) - \beta_i \right] \tag{10}$$

## C.1   Learning values

We begin an auxiliary lemma. Some of the analysis in this section is adapted from [14] which studies Q-learning under the single agent MDP setting.

**Lemma 11.** *([14, Lemma 4.1]) The following properties hold for $\alpha_t^i$:*

1. $\frac{1}{\sqrt{t}} \leq \sum_{i=1}^t \frac{\alpha_t^i}{\sqrt{i}} \leq \frac{2}{\sqrt{t}}$ *for every $t \geq 1$.*

2. $\max_{i \in [t]} \alpha_t^i \leq \frac{2H}{t}$ *and* $\sum_{i=1}^t (\alpha_t^i)^2 \leq \frac{2H}{t}$ *for every $t \geq 1$.*

3. $\sum_{t=i}^\infty \alpha_t^i = 1 + \frac{1}{H}$ *for every $i \geq 1$.*

We also define $\tilde{\beta}_t := 2\sum_{i=1}^t \alpha_t^i \beta_i \leq \mathcal{O}(\sqrt{H^3\iota/t})$. Now we are ready to prove Lemma 3.

*Proof of Lemma 3.* We give the proof for one direction and the other direction is similar. For the proof of the first claim, let $t = N_h^k(s,a,b)$ and suppose $(s,a,b)$ was previously taken at episodes $k^1, \ldots, k^t < k$ at the $h$-th step. Let $\mathcal{F}_i$ be the $\sigma$-algebra generated by all the random variables in until the $k^i$-th episode. Then $\{\alpha_t^i[(\widehat{\mathbb{P}}_h^{k^i} - \mathbb{P}_h)V_{h+1}^\star](s,a,b)\}_{i=1}^t$ is a martingale differene sequence w.r.t. the filtration $\{\mathcal{F}_i\}_{i=1}^t$. By Azuma-Hoeffding,

$$\left| \sum_{i=1}^t \alpha_t^i \left[ \left( \widehat{\mathbb{P}}_h^{k^i} - \mathbb{P}_h \right) V_{h+1}^\star \right](s,a,b) \right| \leq 2H \sqrt{\sum_{i=1}^t \left( \alpha_t^i \right)^2 \iota} \leq \tilde{\beta}_t$$

Here we prove a stronger version of the first claim by induction: for any $(s,a,b,h,k) \in \mathcal{S} \times \mathcal{A} \times \mathcal{B} \times [H] \times [K]$,

$$\overline{Q}_h^k(s,a,b) \geq Q_h^\star(s,a,b) \geq \underline{Q}_h^k(s,a,b), \quad \overline{V}_h^k(s) \geq V_h^\star(s) \geq \underline{V}_h^k(s).$$

Suppose the guarantee is true for $h + 1$, then by the above concentration result,

$$(\overline{Q}_h^k - Q_h^\star)(s, a, b) \geq \alpha_t^0 H + \sum_{i=1}^t \alpha_t^i \left(\overline{V}_{h+1}^{k^i} - V_{h+1}^\star\right)\left(s_{h+1}^{k^i}\right) \geq 0.$$

Also,

$$\overline{V}_h^k(s) - V_h^\star(s) = (\mathbb{D}_{\pi_h^k}\overline{Q}_{h+1}^k)(s) - \max_{\mu \in \Delta_\mathcal{A}} \min_{\nu \in \Delta_\mathcal{B}} (\mathbb{D}_{\mu \times \nu} Q_{h+1}^\star)(s)$$

$$\geq \max_{\mu \in \Delta_\mathcal{A}} (\mathbb{D}_{\mu \times \nu_h^k} \overline{Q}_h^k)(s) - \max_{\mu \in \Delta_\mathcal{A}} (\mathbb{D}_{\mu \times \nu_h^k} Q_h^\star)(s) \geq 0$$

where $\overline{Q}_h^k(s, a, b) \geq Q_h^\star(s, a, b)$ has just been proved. The other direction is proved similarly.

Now we continue with the proof of the second claim. Let $t = n_h^k$ and define $\delta_h^k := \left(\overline{V}_h^k - \underline{V}_h^k\right)\left(s_h^k\right)$, then by definition

$$\delta_h^k = \mathbb{E}_{(a,b) \sim \pi_h^k} \left(\overline{Q}_h^k - \underline{Q}_h^k\right)\left(s_h^k, a, b\right) = \left(\overline{Q}_h^k - \underline{Q}_h^k\right)\left(s_h^k, a_h^k, b_h^k\right) + \zeta_h^k$$

$$\overset{(i)}{=} \alpha_t^0 H + \sum_{i=1}^t \alpha_t^i \delta_{h+1}^{k_h^i(s_h^k, a_h^k, b_h^k)} + 2\tilde{\beta}_t + \zeta_h^k$$

where $(i)$ is by taking the difference of equation (9) and equation (10) and

$$\zeta_h^k := \mathbb{E}_{(a,b) \sim \pi_h^k} \left(\overline{Q}_h^k - \underline{Q}_h^k\right)\left(s_h^k, a, b\right) - \left(\overline{Q}_h^k - \underline{Q}_h^k\right)\left(s_h^k, a_h^k, b_h^k\right)$$

is a martingale difference sequence.

Taking the summation w.r.t. $k$, we begin with the first two terms,

$$\sum_{k=1}^K \alpha_{n_h^k}^0 H = \sum_{k=1}^K H \mathbb{I}\left\{n_h^k = 0\right\} \leq SABH$$

$$\sum_{k=1}^K \sum_{i=1}^{n_h^k} \alpha_{n_h^k}^i \delta_{h+1}^{k_h^i\left(s_h^k, a_h^k, b_h^k\right)} \overset{(i)}{\leq} \sum_{k'=1}^K \delta_{h+1}^{k'} \sum_{i=n_h^{k'}+1}^\infty \alpha_i^{n_h^{k'}} \overset{(ii)}{\leq} \left(1 + \frac{1}{H}\right) \sum_{k=1}^K \delta_{h+1}^k.$$

where $(i)$ is by changing the order of summation and $(ii)$ is by Lemma 11.

Plugging them in,

$$\sum_{k=1}^K \delta_h^k \leq SABH + \left(1 + \frac{1}{H}\right) \sum_{k=1}^K \delta_{h+1}^k + \sum_{k=1}^K \left(2\tilde{\beta}_{n_h^k} + \zeta_h^k\right).$$

Recursing this argument for $h \in [H]$ gives

$$\sum_{k=1}^K \delta_1^k \leq eSABH^2 + 2e \sum_{h=1}^H \sum_{k=1}^K \tilde{\beta}_{n_h^k} + \sum_{h=1}^H \sum_{k=1}^K (1 + 1/H)^{h-1} \zeta_h^k$$

By pigeonhole argument,

$$\sum_{k=1}^K \tilde{\beta}_{n_h^k} \leq \mathcal{O}(1) \sum_{k=1}^K \sqrt{\frac{H^3 \iota}{n_h^k}} = \mathcal{O}(1) \sum_{s,a,b}^{N_h^K(s,a,b)} \sum_{n=1} \sqrt{\frac{H^3 \iota}{n}} \leq \mathcal{O}\left(\sqrt{H^3 SABK\iota}\right) = \mathcal{O}\left(\sqrt{H^2 SABT\iota}\right)$$

By Azuma-Hoeffding,

$$\sum_{h=1}^H \sum_{k=1}^K (1 + 1/H)^{h-1} \zeta_h^k \leq e\sqrt{2H^3 K\iota} = eH\sqrt{2T\iota}$$

with high probability. The proof is completed by putting everything together.

$\square$

## C.2 Certified policies

Algorithm 1 only learns the value of game but itself cannot give a near optimal policy for each player. In this section, we analyze the certified policy based on the above exploration process (Algorithm 2) and prove the sample complexity guarantee. To this end, we need to first define a new group of policies $\hat{\mu}_h^k$ to facilitate the proof , and $\hat{\nu}_h^k$ are defined similarly. Notice $\hat{\mu}_h^k$ is related to $\hat{\mu}$ defined in Algorithm 2 by $\hat{\mu} = \frac{1}{k} \sum_{i=1}^k \hat{\mu}_1^i$.

---

**Algorithm 5** Policy $\hat{\mu}_h^k$

1: **Initialize:** $k' \leftarrow k$.
2: **for** step $h' = h, h+1, \ldots, H$ **do**
3:     Observe $s_{h'}$.
4:     Sample $a_{h'} \sim \mu_h^{k'}(s_{h'})$.
5:     Observe $b_{h'}$.
6:     $t \leftarrow N_h^{k'}(s_{h'}, a_{h'}, b_{h'})$.
7:     Sample $i$ from $[t]$ with $\mathbb{P}(i) = \alpha_t^i$.
8:     $k' \leftarrow k_{h'}^i(s_{h'}, a_{h'}, b_{h'})$

---

We also define $\hat{\mu}_{h+1}^k[s, a, b]$ for $h \leq H - 1$, which is an intermediate algorithm only involved in the analysis. The above two policies are related by $\hat{\mu}_{h+1}^k[s, a, b] = \sum_{i=1}^t \alpha_t^i \hat{\mu}_{h+1}^k$ where $t = N_h^k(s, a, b)$. $\hat{\nu}_{h+1}^k[s, a, b]$ is defined similarly.

---

**Algorithm 6** Policy $\hat{\mu}_{h+1}^k[s, a, b]$

1: $t \leftarrow N_h^k(s, a, b)$.
2: Sample $i$ from $[t]$ with $\mathbb{P}(i) = \alpha_t^i$.
3: $k' \leftarrow k_h^i(s, a, b)$
4: **for** step $h' = h+1, \ldots, H$ **do**
5:     Observe $s_{h'}$.
6:     Sample $a_{h'} \sim \mu_h^{k'}(s_{h'})$.
7:     Observe $b_{h'}$.
8:     $t \leftarrow N_h^{k'}(s_{h'}, a_{h'}, b_{h'})$.
9:     Sample $i$ from $[t]$ with $\mathbb{P}(i) = \alpha_t^i$.
10:     $k' \leftarrow k_{h'}^i(s_{h'}, a_{h'}, b_{h'})$

---

Since the policies defined in Algorithm 5 and Algorithm 6 are non-Markov, many notations for values of Markov policies are no longer valid here. To this end, we need to define the value and Q-value of general policies starting from step $h$, if the general policies starting from the $h$-th step do not depends the history before the $h$-th step. Notice the special case $h = 1$ has already been covered in Section 2. For a pair of general policy $(\mu, \nu)$ which *does not depend* on the hostory before the $h$-th step, we can still use the same definitions (1) and (2) to define their value $V_h^{\mu,\nu}(s)$ and $Q_h^{\mu,\nu}(s, a, b)$ at step $h$. We can also define the best response $\nu^\dagger(\mu)$ of a general policy $\mu$ as the minimizing policy so that $V_h^{\mu,\dagger}(s) \equiv V_h^{\mu,\nu^\dagger(\mu)}(s) = \inf_\nu V_h^{\mu,\nu}(s)$ at step $h$. Similarly, we can define $Q_h^{\mu,\dagger}(s, a, b) \equiv Q_h^{\mu,\nu^\dagger(\mu)}(s, a, b) = \inf_\nu Q_h^{\mu,\nu}(s, a, b)$. As before, the best reponse of a general policy is not necessarily Markovian.

It should be clear from the definition of Algorithm 5 and Algorithm 6 that $\hat{\mu}_h^k$, $\hat{\nu}_h^k$, $\hat{\mu}_{h+1}^k[s, a, b]$ and $\hat{\nu}_{h+1}^k[s, a, b]$ does not depend on the history before step $h$, therefore related value and Q-value functions are well defined for the corresponding steps. Now we can show the policies defined above are indeed certified.

**Lemma 12.** *For any $p \in (0, 1)$, with probability at least $1 - p$, the following holds for any* $(s, a, b, h, k) \in \mathcal{S} \times \mathcal{A} \times \mathcal{B} \times [H] \times [K]$,

$$\overline{Q}_h^k(s, a, b) \geq Q_h^{\dagger, \hat{\nu}_{h+1}^k[s,a,b]}(s, a, b), \quad \overline{V}_h^k(s) \geq V_h^{\dagger, \hat{\nu}_h^k}(s)$$

$$\underline{Q}_h^k(s, a, b) \leq Q_h^{\hat{\mu}_{h+1}^k[s,a,b], \dagger}(s, a, b), \quad \underline{V}_h^k(s) \leq V_h^{\hat{\mu}_h^k, \dagger}(s)$$

*Proof of Lemma 12.* We first prove this for $h = H$.

$$\overline{Q}_H^k(s,a,b) = \alpha_t^0 H + \sum_{i=1}^{t} \alpha_t^i \left[ r_H(s,a,b) + \beta_i \right]$$

$$\geq r_H(s,a,b) = Q_H^{\dagger, \hat{\nu}_{H+1}^k}(s,a,b)$$

because $H$ is the last step and

$$\overline{V}_H^k(s) = (\mathbb{D}_{\pi_H^k} \overline{Q}_H^k)(s) \geq \sup_{\mu \in \Delta_{\mathcal{A}}} (\mathbb{D}_{\mu \times \nu_H^k} \overline{Q}_H^k)(s)$$

$$\geq \sup_{\mu \in \Delta_{\mathcal{A}}} (\mathbb{D}_{\mu \times \nu_H^k} r_H)(s) = V_H^{\dagger, \nu_H^k}(s) = V_H^{\dagger, \hat{\nu}_H^k}(s)$$

because $\pi_H^k$ is CCE, and by definition $\hat{\nu}_H^k = \nu_H^k$.

Now suppose the claim is true for $h + 1$, consider the $h$ case. Consider a fixed tuple $(s, a, b)$ and let $t = N_h^k(s, a, b)$. Suppose $(s, a, b)$ was previously taken at episodes $k^1, \ldots, k^t < k$ at the $h$-th step. Let $\mathcal{F}_i$ be the $\sigma$-algebra generated by all the random variables in until the $k^i$-th episode. Then $\{\alpha_t^i [r_h(s,a,b) + V_{h+1}^{\dagger, \hat{\nu}_{h+1}^{k^i}}(s_{h+1}^{k^i}) + \beta_i]\}_{i=1}^{t}$ is a martingale differene sequence w.r.t. the filtration $\{\mathcal{F}_i\}_{i=1}^{t}$. By Azuma-Hoeffding and the definition of $b_i$,

$$\sum_{i=1}^{t} \alpha_t^i \left[ r_h(s,a,b) + V_{h+1}^{\dagger, \hat{\nu}_{h+1}^{k^i}}(s_{h+1}^{k^i}) + \beta_i \right] \geq \sum_{i=1}^{t} \alpha_t^i Q_h^{\dagger, \hat{\nu}_{h+1}^{k^i}}(s,a,b)$$

with high probability. Combining this with the induction hypothesis,

$$\overline{Q}_h^k(s,a,b) = \alpha_t^0 H + \sum_{i=1}^{t} \alpha_t^i \left[ r_h(s,a,b) + \overline{V}_{h+1}^{k^i}(s_{h+1}^{k^i}) + \beta_i \right]$$

$$\geq \sum_{i=1}^{t} \alpha_t^i \left[ r_h(s,a,b) + V_{h+1}^{\dagger, \hat{\nu}_{h+1}^{k^i}}(s_{h+1}^{k^i}) + \beta_i \right] \geq \sum_{i=1}^{t} \alpha_t^i Q_h^{\dagger, \hat{\nu}_{h+1}^{k^i}}(s,a,b)$$

$$\overset{(i)}{\geq} \max_{\mu} \sum_{i=1}^{t} \alpha_t^i Q_h^{\mu, \hat{\nu}_{h+1}^{k^i}}(s,a,b) = Q_h^{\dagger, \hat{\nu}_{h+1}^k[s,a,b]}(s,a,b)$$

where we have taken the maximum operator out of the summation in $(i)$, which does not increase the sum.

On the other hand,

$$\overline{V}_h^k(s) = (\mathbb{D}_{\pi_h^k} \overline{Q}_h^k)(s) \overset{(i)}{\geq} \sup_{\mu \in \Delta_{\mathcal{A}}} (\mathbb{D}_{\mu \times \nu_h^k} \overline{Q}_h^k)(s)$$

$$\overset{(ii)}{\geq} \max_{a \in \mathcal{A}} \mathbb{E}_{b \sim \nu_h^k} Q_h^{\dagger, \hat{\nu}_{h+1}^k[s,a,b]}(s,a,b) = V_h^{\dagger, \hat{\nu}_h^k}(s)$$

where $(i)$ is by the definition of CCE and $(ii)$ is the induction hypothesis. The other direction is proved by performing smilar arguments on $\underline{Q}_h^k(s,a,b)$, $Q_h^{\hat{\mu}_{h+1}^k[s,a,b],\dagger}(s,a,b)$, $\underline{V}_h^k(s)$ and $V_h^{\hat{\mu}_h^k,\dagger}(s)$. $\square$

Finally we give the theoretical guarantee of the policies defined above.

*Proof of Theorem 4.* By lemma 12, we have

$$\sum_{k=1}^{K} \left( V_1^{\dagger, \hat{\nu}_1^k} - V_1^{\hat{\mu}_1^k, \dagger} \right)(s_1) \leq \sum_{k=1}^{K} \left( \overline{V}_1^k - \underline{V}_1^k \right)(s_1)$$

and Lemma 3 upper bounds this quantity by

$$\sum_{k=1}^{K} \left( V_1^{\dagger, \hat{\nu}_1^k} - V_1^{\hat{\mu}_1^k, \dagger} \right)(s_1) \le \mathcal{O}\left( \sqrt{H^4 SABT\iota} \right)$$

By definition of the induced policy, with probability at least $1 - p$, if we run Nash Q-learning (Algorithm 1) for $K$ episodes with

$$K \ge \Omega\left( \frac{H^5 SAB\iota}{\epsilon^2} \right),$$

its induced policies $(\hat{\mu}, \hat{\nu})$ (Algorithm 2) will be $\epsilon$-optimal in the sense $V_1^{\dagger, \hat{\nu}}(s_1) - V_1^{\hat{\mu}, \dagger}(s_1) \le \epsilon$. $\quad\square$

# D  Proof for Nash V-learning

In this section, we present proofs of the results in Section 4. We denote $V^k, \mu^k, \nu^k$ for values and policies *at the beginning* of the $k$-th episode. We also introduce the following short-hand notation $[\widehat{\mathbb{P}}_h^k V](s, a, b) := V(s_{h+1}^k)$.

We will use the following notations several times later: suppose the state $s$ was visited at episodes $k^1, k^2, \dots$ at the $h$-th step. Since the definition of $k^i$ depends on the state $s$, we will show the dependence explicitly by writing $k_h^i(s)$ when necessary and omit it when there is no confusion. We also define $N_h^k(s)$ to be the number of times the state $s$ has been visited *at the beginning* of the $k$-th episode. Finally we denote $n_h^k = N_h^k(s_h^k)$. Notice the definitions here are different from that in Appendix C.

The following lemma is a simple consequence of the update rule in Algorithm 3, which will be used several times later.

**Lemma 13.** *Let $t = N_h^k(s)$ and suppose $s$ was previously visited at episodes $k^1, \dots, k^t < k$ at the $h$-th step. The update rule in Algorithm 3 is equivalent to the following equations.*

$$\overline{V}_h^k(s) = \alpha_t^0 H + \sum_{i=1}^{t} \alpha_t^i \left[ r_h(s, a_h^{k^i}, b_h^{k^i}) + \overline{V}_{h+1}^{k^i}(s_{h+1}^{k^i}) + \overline{\beta}_i \right] \tag{11}$$

$$\underline{V}_h^k(s) = \sum_{i=1}^{t} \alpha_t^i \left[ r_h(s, a_h^{k_h^i}, b_h^{k_h^i}) + \underline{V}_{h+1}^{k_h^i}(s_{h+1}^{k_h^i}) - \underline{\beta}_i \right] \tag{12}$$

## D.1  Missing algorithm details

We first give Algorithm 7: the min-player counterpart of Algorithm 3. Almost everything is symmetric except the definition of loss function to keep it non-negative.

## D.2  Learning values

As usual, we begin with learning the value $V^\star$ of the Markov game. We begin with an auxiliary lemma, which justifies our choice of confidence bound.

**Lemma 14.** *Let $t = N_h^k(s)$ and suppose state $s$ was previously taken at episodes $k^1, \dots, k^t < k$ at the $h$-th step. Choosing $\overline{\eta}_t = \sqrt{\frac{\log A}{At}}$ and $\underline{\eta}_t = \sqrt{\frac{\log B}{Bt}}$, with probability $1 - p$, for any $(s, h, t) \in \mathcal{S} \times [H] \times [K]$, there exist a constant $c$ s.t.*

$$\max_{\mu} \sum_{i=1}^{t} \alpha_t^i \mathbb{D}_{\mu \times \nu_h^{k^i}} \left( r_h + \mathbb{P}_h \overline{V}_{h+1}^{k^i} \right)(s) - \sum_{i=1}^{t} \alpha_t^i \left[ r_h\left(s, a_h^{k^i}, b_h^{k^i}\right) + \overline{V}_{h+1}^{k^i}\left(s_{h+1}^{k^i}\right) \right] \le c\sqrt{2H^4 A\iota/t}$$

$$\sum_{i=1}^{t} \alpha_t^i \left[ r_h\left(s, a_h^{k^i}, b_h^{k^i}\right) + \underline{V}_{h+1}^{k^i}\left(s_{h+1}^{k^i}\right) \right] - \min_{\nu} \sum_{i=1}^{t} \alpha_t^i \mathbb{D}_{\mu_h^{k^i} \times \nu} \left( r_h + \mathbb{P}_h \overline{V}_{h+1}^{k^i} \right)(s) \le c\sqrt{2H^4 B\iota/t}$$

---

**Algorithm 7** Optimistic Nash V-learning (the min-player version)

---

1: **Initialize:** for any $(s, a, b, h)$, $\underline{V}_h(s) \leftarrow 0$, $\underline{L}_h(s, b) \leftarrow 0$, $N_h(s) \leftarrow 0$, $\nu_h(b|s) \leftarrow 1/B$.
2: **for** episode $k = 1, \ldots, K$ **do**
3:     receive $s_1$.
4:     **for** step $h = 1, \ldots, H$ **do**
5:         take action $b_h \sim \nu_h(\cdot|s_h)$, observe the action $a_h$ from opponent
6:         observe reward $r_h(s_h, a_h, b_h)$ and next state $s_{h+1}$.
7:         $t = N_h(s_h) \leftarrow N_h(s_h) + 1$.
8:         $\underline{V}_h(s_h) \leftarrow \max\{0, (1 - \alpha_t)\underline{V}_h(s_h) + \alpha_t(r_h(s_h, a_h, b_h) + \underline{V}_{h+1}(s_{h+1}) - \underline{\beta}_t)\}$
9:         **for** all $b \in \mathcal{B}$ **do**
10:            $\ell_h(s_h, b) \leftarrow [r_h(s_h, a_h, b_h) + \underline{V}_{h+1}(s_{h+1})]\mathbb{I}\{b_h = b\}/[\nu_h(b_h|s_h) + \underline{\eta}_t]$.
11:            $\underline{L}_h(s_h, b) \leftarrow (1 - \alpha_t)\underline{L}_h(s_h, b) + \alpha_t \ell_h(s_h, b)$.
12:         set $\nu_h(\cdot|s_h) \propto \exp[-(\underline{\eta}_t/\alpha_t)\underline{L}_h(s_h, \cdot)]$.

---

*Proof of Lemma 14.* We prove the first inequality. The proof for the second inequality is similar. We consider thoughout the proof a fixed $(s, h, t) \in \mathcal{S} \times [H] \times [K]$. Define $\mathcal{F}_i$ as the $\sigma$-algebra generated by all the random variables before the $k_h^i$-th episode. Then $\{r_h(s, a_h^{k^i}, b_h^{k^i}) + \overline{V}_{h+1}^{k^i}(s_{h+1}^{k^i})\}_{i=1}^t$ is a martingale sequence w.r.t. the filtration $\{\mathcal{F}_i\}_{i=1}^t$. By Azuma-Hoeffding,

$$\sum_{i=1}^t \alpha_t^i \mathbb{D}_{\mu_h^{k^i} \times \nu_h^{k^i}} \left( r_h + \mathbb{P}_h \overline{V}_{h+1}^{k^i} \right)(s) - \sum_{i=1}^t \alpha_t^i \left[ r_h \left( s, a_h^{k^i}, b_h^{k^i} \right) + \overline{V}_{h+1}^{k^i} \left( s_{h+1}^{k^i} \right) \right] \leq 2\sqrt{H^3 \iota / t}$$

So we only need to bound

$$\max_\mu \sum_{i=1}^t \alpha_t^i \mathbb{D}_{\mu \times \nu_h^{k^i}} \left( r_h + \mathbb{P}_h \overline{V}_{h+1}^{k^i} \right)(s) - \sum_{i=1}^t \alpha_t^i \mathbb{D}_{\mu_h^{k^i} \times \nu_h^{k^i}} \left( r_h + \mathbb{P}_h \overline{V}_{h+1}^{k^i} \right)(s) := R_t^\star \quad (13)$$

where $R_t^\star$ is the weighted regret in the first $t$ times of visiting state $s$, with respect to the optimal policy in hindsight, in the following adversarial bandit problem. The loss function is defined by

$$l_i(a) = \mathbb{E}_{b \sim \nu_h^{k^i}(s)} \{ H - h + 1 - r_h(s, a, b) - \mathbb{P}_h \overline{V}_{h+1}^{k^i}(s, a, b) \}$$

with weight $w_i = \alpha_t^i$. We note the weighted regret can be rewrite as $R_t^\star = \sum_{i=1}^t w_i \left\langle \mu_h^\star - \mu_h^{k_i}, l_i \right\rangle$ where $\mu_h^\star$ is argmax for (13), and the loss function satisfies $l_i(a) \in [0, H]$

Therefore, Algorithm 3 is essentially performing follow the regularized leader (FTRL) algorithm with changing step size for each state to solve this adversarial bandit problem. The policy we are using is $\mu_h^{k^i}(s, a)$ and the optimistic biased estimator

$$\hat{l}_i(a) = \frac{H - h + 1 - r_h(s_h^{k^i}, a_h^{k^i}, b_h^{k^i}) - \overline{V}_{h+1}^{k^i}(s_{h+1}^{k^i})}{\mu_h^{k^i}(s, a) + \overline{\eta}_i} \cdot \mathbb{I}\left\{ a_h^{k^i} = a \right\}$$

is used to handle the bandit feedback.

A more detailed discussion on how to solve the weighted adversarial bandit problem is included in Appendix F. Note that $w_i = \alpha_t^i$ is monotonic inscreasing, i.e. $\max_{i \leq t} w_i = w_t$. By Lemma 17, we have

$$R_t^\star \leq 2H\alpha_t^t \sqrt{At\iota} + \frac{3H\sqrt{A\iota}}{2} \sum_{i=1}^t \frac{\alpha_t^i}{\sqrt{i}} + \frac{1}{2} H\alpha_t^t \iota + H \sqrt{2\iota \sum_{i=1}^t (\alpha_t^i)^2}$$

$$\leq 4H^2\sqrt{A\iota/t} + 3H\sqrt{A\iota/t} + H^2\iota/t + \sqrt{4H^3\iota/t}$$

$$\leq 10H^2\sqrt{A\iota/t}$$

with probability $1 - p/(SHK)$. Finally by a union bound over all $(s, h, t) \in \mathcal{S} \times [H] \times [K]$, we finish the proof. $\qquad\square$

We now prove the following Lemma 15, which is an analogue of Lemma 3 in Nash Q-learning.

**Lemma 15.** *For any $p \in (0,1]$, choose hyperparameters as in (6) for large absolute constant $c$ and $\iota = \log(SABT/p)$. Then, with probability at least $1 - p$, Algorithm 3 and 7 will jointly provide the following guarantees*

- $\overline{V}_h^k(s) \geq V_h^\star(s) \geq \underline{V}_h^k(s)$ for all $(s, h, k) \in \mathcal{S} \times [K] \times [H]$.

- $(1/K) \cdot \sum_{k=1}^K (\overline{V}_1^k - \underline{V}_1^k)(s_1) \leq \mathcal{O}\left(\sqrt{H^6 S(A+B)\iota/K}\right)$.

*Proof of Lemma 15.* We proof the first claim by backward induction. The claim is true for $h = H + 1$. Asumme for any s, $\overline{V}_{h+1}^k(s) \geq V_{h+1}^\star(s)$, $\underline{V}_{h+1}^k(s) \leq V_{h+1}^\star(s)$. For a fixed $(s, h) \in \mathcal{S} \times [H]$ and episode $k \in [K]$, let $t = N_h^k(s)$ and suppose $s$ was previously visited at episodes $k^1, \ldots, k^t < k$ at the $h$-th step. By Bellman equation,

$$
\begin{aligned}
V_h^\star(s) =& \max_\mu \min_\nu \mathbb{D}_{\mu \times \nu}\left(r_h + \mathbb{P}_h V_{h+1}^\star\right)(s) \\
=& \max_\mu \sum_{i=1}^t \alpha_t^i \min_\nu \mathbb{D}_{\mu \times \nu}\left(r_h + \mathbb{P}_h V_{h+1}^\star\right)(s) \\
\leq& \max_\mu \sum_{i=1}^t \alpha_t^i \mathbb{D}_{\mu \times \nu_h^{k^i}}\left(r_h + \mathbb{P}_h V_{h+1}^\star\right)(s) \\
\leq& \max_\mu \sum_{i=1}^t \alpha_t^i \mathbb{D}_{\mu \times \nu_h^{k^i}}\left(r_h + \mathbb{P}_h \overline{V}_{h+1}^{k^i}\right)(s)
\end{aligned}
$$

Comparing with the decomposition of $\overline{V}_h^k(s)$ in Equation (11) and use Lemma 14, we can see if $\overline{\beta}_t = c\sqrt{AH^4\iota/t}$, then $\overline{V}_h^k(s) \geq V_h^\star(s)$. Similar by taking $\underline{\beta}_t = c\sqrt{BH^4\iota/t}$, we also have $\underline{V}_h^k(s) \leq V_h^\star(s)$.

The second cliam is to bound $\delta_h^k := \overline{V}_h^k(s_h^k) - \underline{V}_h^k(s_h^k) \geq 0$. Similar to what we have done in Nash Q-learning analysis, taking the difference of Equation (11) and Equation (12),

$$
\begin{aligned}
\delta_h^k =& \overline{V}_h^k(s_h^k) - \underline{V}_h^k(s_h^k) \\
=& \alpha_{n_h^k}^0 H + \sum_{i=1}^{n_h^k} \alpha_{n_h^k}^i \left[\left(\overline{V}_{h+1}^{k_h^i(s_h^k)} - \underline{V}_{h+1}^{k_h^i(s_h^k)}\right)\left(s_{h+1}^{k_h^i(s_h^k)}\right) + \overline{\beta}_i + \underline{\beta}_i\right] \\
=& \alpha_{n_h^k}^0 H + \sum_{i=1}^{n_h^k} \alpha_{n_h^k}^i \delta_{h+1}^{k_h^i(s_h^k)} + \tilde{\beta}_{n_h^k}
\end{aligned}
$$

where

$$
\tilde{\beta}_j := \sum_{i=1}^j \alpha_j^i(\overline{b}_i + \underline{b}_i) \leq c\sqrt{(A+B)H^4\iota/j}.
$$

Taking the summation w.r.t. $k$, we begin with the first two terms,

$$
\sum_{k=1}^K \alpha_{n_h^k}^0 H = \sum_{k=1}^K H\mathbb{I}\left\{n_h^k = 0\right\} \leq SH
$$

$$
\sum_{k=1}^K \sum_{i=1}^{n_h^k} \alpha_{n_h^k}^i \delta_{h+1}^{k_h^i(s_h^k)} \overset{(i)}{\leq} \sum_{k'=1}^K \delta_{h+1}^{k'} \sum_{i=n_h^{k'}+1}^\infty \alpha_i^{n_h^{k'}} \overset{(ii)}{\leq} \left(1 + \frac{1}{H}\right)\sum_{k=1}^K \delta_{h+1}^k.
$$

where $(i)$ is by changing the order of summation and $(ii)$ is by Lemma 11. Putting them together,

---

**Algorithm 8** Policy $\hat{\mu}_h^k$

---

1: sample $k' \leftarrow \text{Uniform}([k])$.
2: **for** step $h' = h, h+1, \ldots, H$ **do**
3:     observe $s_{h'}$, and set $t \leftarrow N_{h'}^{k'}(s_{h'})$.
4:     sample $m \in [t]$ with $\mathbb{P}(m = i) = \alpha_t^i$.
5:     $k' \leftarrow k_{h'}^m(s_{h'})$.
6:     take action $a_{h'} \sim \mu_{h'}^{k'}(\cdot|s_{h'})$.

---

$$\sum_{k=1}^{K} \delta_h^k = \sum_{k=1}^{K} \alpha_{n_h^k}^0 H + \sum_{k=1}^{K} \sum_{i=1}^{n_h^k} \alpha_{n_h^k}^i \delta_{h+1}^{k_h^i(s_h^k)} + \sum_{k=1}^{K} \tilde{\beta}_{n_h^k}$$

$$\leq HS + \left(1 + \frac{1}{H}\right) \sum_{k=1}^{K} \delta_{h+1}^k + \sum_{k=1}^{K} \tilde{\beta}_{n_h^k}$$

Recursing this argument for $h \in [H]$ gives

$$\sum_{k=1}^{K} \delta_1^k \leq eSH^2 + e \sum_{h=1}^{H} \sum_{k=1}^{K} \tilde{\beta}_{n_h^k}$$

By pigeonhole argument,

$$\sum_{k=1}^{K} \tilde{\beta}_{n_h^k} \leq \mathcal{O}(1) \sum_{k=1}^{K} \sqrt{\frac{(A+B)H^4\iota}{n_h^k}} = \mathcal{O}(1) \sum_{s} \sum_{n=1}^{n_h^K(s)} \sqrt{\frac{(A+B)H^4\iota}{n}}$$

$$\leq \mathcal{O}\left(\sqrt{H^4 S(A+B)K\iota}\right) = \mathcal{O}\left(\sqrt{H^3 S(A+B)T\iota}\right)$$

Expanding this formula repeatedly and apply pigeonhole argument we have

$$\sum_{k=1}^{K} [\overline{V}_h^k - \underline{V}_h^k](s_1) \leq \mathcal{O}(\sqrt{H^5 S(A+B)T\iota}).$$

which finishes the proof. $\qquad\qquad\qquad\qquad\qquad\qquad\qquad\qquad\qquad\qquad\qquad\quad$ $\square$

### D.3   Certified policies

As before, we construct a series of new policies $\hat{\mu}_h^k$ in Algorithm 8. Notice $\hat{\mu}_h^k$ is related to $\hat{\mu}$ defined in Algorithm 4 by $\hat{\mu} = \frac{1}{k} \sum_{i=1}^{k} \hat{\mu}_1^i$. Also we need to consider value and Q-value functions of general policies which *does not depend* on the hostory before the $h$-th step. See Appendix C.2 for details. Again, we can show the policies defined above are indeed certified.

**Lemma 16.** *For any $p \in (0,1)$, with probability at least $1 - p$, the following holds for any $(s, a, b, h, k) \in \mathcal{S} \times \mathcal{A} \times \mathcal{B} \times [H] \times [K]$,*

$$\overline{V}_h^k(s) \geq V_h^{\dagger, \hat{\nu}_h^k}(s), \quad \underline{V}_h^k(s) \leq V_h^{\hat{\mu}_h^k, \dagger}(s)$$

*Proof of Lemma 16.* We prove one side by induction and the other side is similar. The claim is trivially satisfied for $h = H + 1$. Suppose it is ture for $h + 1$, consider a fixed state $s$. Let $t = N_h^k(s)$ and suppose $s$ was previously visited at episodes $k^1, \ldots, k^t < k$ at the $h$-th step. Then using

Lemma 13,

$$\overline{V}_h^k(s) = \alpha_t^0 H + \sum_{i=1}^{t} \alpha_t^i \left[ r_h(s, a_h^{k^i}, b_h^{k_h^i}) + \overline{V}_{h+1}^{k^i}(s_{h+1}^{k^i}) + \overline{\beta}_i \right]$$

$$\overset{(i)}{\geq} \max_{\mu} \sum_{i=1}^{t} \alpha_t^i \mathbb{D}_{\mu \times \nu_h^{k^i}} \left( r_h + \mathbb{P}_h \overline{V}_{h+1}^{k^i} \right)(s)$$

$$\overset{(ii)}{\geq} \max_{\mu} \sum_{i=1}^{t} \alpha_t^i \mathbb{D}_{\mu \times \nu_h^{k^i}} \left( r_h + \mathbb{P}_h V_{h+1}^{\dagger, \hat{\nu}_{h+1}^{k^i}} \right)(s)$$

$$= V_h^{\dagger, \hat{\nu}_h^k}(s)$$

where $(i)$ is by using Lemma 14 and the definition of $\overline{\beta}_i$, and $(ii)$ is by induction hypothesis. $\square$

Equipped with the above lemmas, we are now ready to prove Theorem 5.

*Proof of Theorem 5.* By lemma 16, we have

$$\sum_{k=1}^{K} \left( V_1^{\dagger, \hat{\nu}_1^k} - V_1^{\hat{\mu}_1^k, \dagger} \right)(s_1) \leq \sum_{k=1}^{K} \left( \overline{V}_1^k - \underline{V}_1^k \right)(s_1)$$

and Lemma 15 upper bounds this quantity by

$$\sum_{k=1}^{K} \left( V_1^{\dagger, \hat{\nu}_1^k} - V_1^{\hat{\mu}_1^k, \dagger} \right)(s_1) \leq \mathcal{O}\left( \sqrt{H^5 S(A+B)T\iota} \right)$$

By definition of the induced policy, with probability at least $1 - p$, if we run Nash V-learning (Algorithm 3) for $K$ episodes with

$$K \geq \Omega\left( \frac{H^6 S(A+B)\iota}{\epsilon^2} \right),$$

its induced policies $(\hat{\mu}, \hat{\nu})$ (Algorithm 4) will be $\epsilon$-optimal in the sense $V_1^{\dagger, \hat{\nu}}(s_1) - V_1^{\hat{\mu}, \dagger}(s_1) \leq \epsilon$. $\square$

# E   Proofs of Hardness for Learning the Best Responses

In this section we give the proof of Theorem 6, and Corollary 8. Our proof is inspired by a computational hardness result for adversarial MDPs in [37, Section 4.2], which constructs a family of adversarial MDPs that are computationally as hard as an agnostic parity learning problem.

Section E.1, E.2, E.3 will be devoted to prove Theorem 6, while Corollary 8 is proved in Section E.4. Towards proving Theorem 6, we will:

- (Section E.1) Construct a Markov game.
- (Section E.2) Define a series of problems where a solution in problem implies another.
- (Section E.3) Based on the believed computational hardness of learning paries with noise (Conjecture 7), we conclude that finding the best response of non-Markov policies is computationally hard.

## E.1   Markov game construction

We now describe a Markov game inspired the adversarial MDP in [37, Section 4.2]. We define a Markov game in which we have $2H$ states, $\{i_0, i_1\}_{i=2}^{H}$, $1_0$ (the initial state) and $\perp$ (the terminal state)[4]. In each state the max-player has two actions $a_0$ and $a_1$, while the min-player has two actions

| State/Action | $(a_0, b_0)$ | $(a_0, b_1)$ | $(a_1, b_0)$ | $(a_1, b_1)$ |
|---|---|---|---|---|
| $i_0$ | $(i+1)_0$ | $(i+1)_0$ | $(i+1)_0$ | $(i+1)_1$ |
| $i_1$ | $(i+1)_1$ | $(i+1)_0$ | $(i+1)_1$ | $(i+1)_1$ |

Table 2: Transition kernel of the hard instance.

$b_0$ and $b_1$. The transition kernel is deterministic and the next state for steps $h \leq H - 1$ is defined in Table 2:

At the $H$-th step, i.e. states $H_0$ and $H_1$, the next state is always $\perp$ regardless of the action chosen by both players. The reward function is always 0 except at the $H$-th step. The reward is determined by the action of the min-player, defined by

| State/Action | $(\cdot, b_0)$ | $(\cdot, b_1)$ |
|---|---|---|
| $H_0$ | 1 | 0 |
| $H_1$ | 0 | 1 |

Table 3: Reward of the hard instance.

At the beginning of every episode $k$, both players pick their own policies $\mu_k$ and $\nu_k$, and execute them throughout the episode. The min-player can possibly pick her policy $\nu_k$ adaptive to all the observations in the earlier episodes. The only difference from the standard Markov game protocol is that the actions of the min-player except the last step will be revealed at the beginning of each episode, to match the setting in agnostic learning parities (Problem 2 below). Therefore we are actually considering a easier problem (for the max-player) and the lower bound naturally applies.

### E.2 A series of computationally hard problems

We first introduce a series of problems and then show how the reduction works.

**Problem 1** The max-player $\epsilon$-approximates the best reponse for any general policy $\nu$ in the Markov game defined in Appendix E.1 with probability at least $1/2$, in $\mathrm{poly}(H, 1/\epsilon)$ time.

**Problem 2** Let $x = (x_1, \cdots, x_n)$ be a vector in $\{0,1\}^n$, $T \subseteq [n]$ and $0 < \alpha < 1/2$. The parity of $x$ on $T$ is the boolean function $\phi_T(x) = \oplus_{i \in T} x_i$. In words, $\phi_T(x)$ outputs 0 if the number of ones in the subvector $(x_i)_{i \in T}$ is even and 1 otherwise. A uniform query oracle for this problem is a randomized algorithm that returns a random uniform vector $x$, as well as a noisy classification $f(x)$ which is equal to $\phi_T(x)$ w.p. $\alpha$ and $1 - \phi_T(x)$ w.p. $1 - \alpha$. All examples returned by the oracle are independent. The learning parity with noise problem consists in designing an algorithm with access to the oracle such that,

- **(Problem 2.1)** w.p at least $1/2$, find a (possibly random) function $h : \{0,1\}^n \to \{0,1\}$ satisy $\mathbb{E}_h P_x[h(x) \neq \phi_T(x)] \leq \epsilon$, in $\mathrm{poly}(n, 1/\epsilon)$ time.

- **(Problem 2.2)** w.p at least $1/4$, find $h : \{0,1\}^n \to \{0,1\}$ satisy $P_x[h(x) \neq \phi_T(x)] \leq \epsilon$, in $\mathrm{poly}(n, 1/\epsilon)$ time.

- **(Problem 2.3)** w.p at least $1 - p$, find $h : \{0,1\}^n \to \{0,1\}$ satisy $P_x[h(x) \neq \phi_T(x)] \leq \epsilon$, in $\mathrm{poly}(n, 1/\epsilon, 1/p)$ time.

We remark that Problem 2.3 is the formal definition of learning parity with noise [20, Definition 2], which is conjectured to be computationally hard in the community (see also Conjecture 7).

**Problem 2.3 reduces to Problem 2.2** Step 1: Repeatly apply algorithm for Problem 2.2 $\ell$ times to get $h_1, \ldots, h_\ell$ such that $\min_i P_x[h_i(x) \neq \phi_T(x)] \leq \epsilon$ with probability at least $1 - (3/4)^\ell$. This costs $\mathrm{poly}(n, \ell, 1/\epsilon)$ time. Let $i_\star = \arg\min_i \mathrm{err}_i$ where $\mathrm{err}_i = P_x[h_i(x) \neq \phi_T(x)]$.

Step 2: Construct estimators using $N$ additional data $(x^{(j)}, y^{(j)})_{j=1}^{N}$,

$$\hat{\text{err}}_i := \frac{\frac{1}{N}\sum_{j=1}^{N}\mathbb{I}\{h_i(x^{(j)}) \neq y^{(j)}\} - \alpha}{1 - 2\alpha}.$$

Pick $\hat{i} = \text{argmin}_i \hat{\text{err}}_i$. When $N \geq \log(1/p)/\epsilon^2$, with probability at least $1 - p/2$, we have

$$\max_i |\hat{\text{err}}_i - \text{err}_i| \leq \frac{\epsilon}{1 - 2\alpha}.$$

This means that

$$\text{err}_{\hat{i}} \leq \hat{\text{err}}_{\hat{i}} + \frac{\epsilon}{1 - 2\alpha} \leq \hat{\text{err}}_{i_\star} + \frac{\epsilon}{1 - 2\alpha} \leq \text{err}_{i_\star} + \frac{2\epsilon}{1 - 2\alpha} \leq O(1)\epsilon.$$

This step uses $\text{poly}(n, N, \ell) = \text{poly}(n, 1/\epsilon, \log(1/p), \ell)$ time.

Step 3: Pick $\ell = \log(1/p)$, we are guaranteed that good events in step 1 and step 2 happen with probability $\geq 1 - p/2$ and altogether happen with probability at least $1 - p$. The total time used is $\text{poly}(n, 1/\epsilon, \log(1/p))$. Note better dependence on $p$ than required.

**Problem 2.2 reduces to Problem 2.1:** If we have an algorithm that gives $\mathbb{E}_{h \sim \mathcal{D}} P_x[h(x) \neq \phi_T(x)] \leq \epsilon$ with probability $1/2$. Then if we sample $\hat{h} \sim \mathcal{D}$, by Markov's inequality, we have with probability $\geq 1/4$ that

$$P_x[\hat{h}(x) \neq \phi_T(x)] \leq 2\epsilon$$

**Problem 2.1 reduces to Problem 1:** Consider the Markov game constructed above with $H - 1 = n$. The only missing piece we fill up here is the policy $\nu$ of the min-player, which is constructed as following. The min-player draws a sample $(x, y)$ from the uniform query oracle, then taking action $b_0$ at the step $h \leq H - 1$ if $x_h = 0$ and $b_1$ otherwise. For the $H$-th step, the min-player take action $b_0$ if $y = 0$ and $b_1$ otherwise. Also notice the policy $\hat{\mu}$ of the max-player can be descibed by a set $\hat{T} \subseteq [H]$ where he takes action $a_1$ at step $h$ if $h$ and $a_0$ otherwise. As a result, the max-player receive non-zero result iff $\phi_{\hat{T}}(x) = y$.

In the Markov game, we have $V_1^{\hat{\mu},\nu}(s_1) = \mathbb{P}(\phi_{\hat{T}}(x) = y)$. As a result, the optimal policy $\mu^*$ corresponds to the true parity set $T$. As a result,

$$(V_1^{\dagger,\nu} - V_1^{\hat{\mu},\nu})(s_1) = \mathbb{P}_{x,y}(\phi_T(x) = y) - \mathbb{P}_{x,y}(\phi_{\hat{T}}(x) = y) \leq \epsilon$$

by the $\epsilon$-approximation guarantee.

Also notice

$$\mathbb{P}_{x,y}(\phi_{\hat{T}}(x) \neq y) - \mathbb{P}_{x,y}(\phi_T(x) \neq y) = (1-\alpha)\mathbb{P}_x(\phi_{\hat{T}}(x) \neq \phi_T(x)) + \alpha\mathbb{P}_x(\phi_{\hat{T}}(x) = \phi_T(x)) - \alpha$$
$$= (1 - 2\alpha)\mathbb{P}_x(\phi_{\hat{T}}(x) \neq \phi_T(x))$$

This implies:

$$\mathbb{P}_x(\phi_{\hat{T}}(x) \neq \phi_T(x)) \leq \frac{\epsilon}{1 - 2\alpha}$$

### E.3  Putting them together

So far, we have proved that Solving Problem 1 implies solving Problem 2.3, where Problem 1 is the problem of learning $\epsilon$-approximate best response in Markov games (the problem we are interested in), and Problem 2.3 is precisely the problem of learning parity with noise [20]. This concludes the proof.

### E.4  Proofs of Hardness Against Adversarial Opponents

Corollary 8 is a direct consequence of Theorem 6, as we will show now.

*Proof of Corollary 8.* We only need to prove a polynomial time no-regret algorithm also learns the best response in a Markov game where the min-player following non-Markov policy $\nu$. Then the no-regret guarantee implies,

$$V_1^{\dagger,\nu}(s_1) - \frac{1}{K}\sum_{k=1}^{K} V_1^{\mu^k,\nu}(s_1) \le \text{poly}(S,H,A,B)K^{-\delta}$$

where $\mu_k$ is the policy of the max-player in the $k$-th episode. If we choose $\hat{\mu}$ uniformly randomly from $\{\mu_k\}_{k=1}^{K}$, then

$$V_1^{\dagger,\nu}(s_1) - V_1^{\hat{\mu},\nu}(s_1) \le \text{poly}(S,H,A,B)K^{-\delta}.$$

Choosing $\epsilon = \text{poly}(S,H,A,B)K^{-\delta}$, $K = \text{poly}(S,H,A,B,1/\epsilon)$ and the running time of the no-regret algorithm is still $\text{poly}(S,H,A,B,1/\epsilon)$ to learn the $\epsilon$-approximate best response.

To see that the Corollary 8 remains to hold for policies that are Markovian in each episode and non-adaptive, we can take the hard instance in Theorem 6 and let $\nu^k$ denote the min-player's policy in the $k$-th episode. Note that each $\nu^k$ is Markovian and non-adaptive on the observations in previous episodes. If there is a polynomial time no-regret algorithm against such $\{\nu^k\}$, then by the online-to-batch conversion similar as the above, the mixture of $\{\mu_k\}_{k=1}^{K}$ learns a best response against $\nu$ in polynomial time.

$\square$

# F    Auxiliary Lemmas for Weighted Adversarial Bandit

In this section, we formulate the bandit problem we reduced to in the proof of Lemma 14. Although the machnisms are already well understood, we did not find a good reference of Follow the Regularized Leader (FTRL) algorithm with

1. changing step size

2. weighted regret

3. high probability regret bound

For completeness, we give the detailed derivation here.

---
**Algorithm 9** FTRL for Weighted Regret with Changing Step Size
---
1: **for** episode $t = 1,\dots,K$ **do**
2:    $\theta_t(a) \propto \exp[-(\eta_t/w_t)\cdot\sum_{i=1}^{t-1} w_i\hat{l}_i(a)]$
3:    Take action $a_t \sim \theta_t(\cdot)$, and observe loss $\tilde{l}_t(a_t)$.
4:    $\hat{l}_t(a) \leftarrow \tilde{l}_t(a)\mathbb{I}\{a_t = a\}/(\theta_t(a) + \gamma_t)$ for all $a \in \mathcal{A}$.
---

We assume $\tilde{l}_i \in [0,1]^A$ and $\mathbb{E}_i\tilde{l}_i = l_i$. Define $A = |\mathcal{A}|$, we set the hyperparameters by

$$\eta_t = \gamma_t = \sqrt{\frac{\log A}{At}}$$

Define the filtration $\mathcal{F}_t$ by the $\sigma$-algebra generated by $\{a_i, l_i\}_{i=1}^{t-1}$. Then the regret can be defined as

$$R_t(\theta^*) := \sum_{i=1}^{t} w_i\mathbb{E}_{a\sim\theta^*}[l_i(a) - l_i(a_i)|\mathcal{F}_i] = \sum_{i=1}^{t} w_i\langle\theta_i - \theta^*, l_i\rangle$$

We can easily check the definitions here is just an abstract version of that in the proof of Lemma 14 with rescaling. To state the regret guarantee, we also define $\iota = \log(p/AK)$ for any $p \in (0,1]$. Now we can upper bound the regret by

**Lemma 17.** *Following Algorithm 9, with probability $1 - 3p$, for any $\theta^* \in \Delta^A$ and $t \leq K$ we have*

$$R_t\left(\theta^*\right) \leq 2 \max_{i \leq t} w_i \sqrt{A t \iota} + \frac{3\sqrt{A\iota}}{2} \sum_{i=1}^{t} \frac{w_i}{\sqrt{i}} + \frac{1}{2} \max_{i \leq t} w_i \iota + \sqrt{2\iota \sum_{i=1}^{t} w_i^2}$$

*Proof.* The regret $R_t(\theta^*)$ can be decomposed into three terms

$$
\begin{aligned}
R_t\left(\theta^*\right) &= \sum_{i=1}^{t} w_i \left\langle \theta_i - \theta^*, l_i \right\rangle \\
&= \underbrace{\sum_{i=1}^{t} w_i \left\langle \theta_i - \theta^*, \hat{l}_i \right\rangle}_{(A)} + \underbrace{\sum_{i=1}^{t} w_i \left\langle \theta_i, l_i - \hat{l}_i \right\rangle}_{(B)} + \underbrace{\sum_{i=1}^{t} w_i \left\langle \theta^*, \hat{l}_i - l_i \right\rangle}_{(C)}
\end{aligned}
$$

and we bound $(A)$ in Lemma 19, $(B)$ in Lemma 20 and $(C)$ in Lemma 21.

Setting $\eta_t = \gamma_t = \sqrt{\frac{\log A}{At}}$, the conditions in Lemma 19 and Lemma 21 are satisfied. Putting them together and take union bound, we have with probability $1 - 3p$

$$R_t\left(\theta^*\right) \leq \frac{w_t \log A}{\eta_t} + \frac{A}{2} \sum_{i=1}^{t} \eta_i w_i + \frac{1}{2} \max_{i \leq t} w_i \iota + A \sum_{i=1}^{t} \gamma_i w_i + \sqrt{2\iota \sum_{i=1}^{t} w_i^2} + \max_{i \leq t} w_i \iota / \gamma_t$$

$$\leq 2 \max_{i \leq t} w_i \sqrt{A t \iota} + \frac{3\sqrt{A\iota}}{2} \sum_{i=1}^{t} \frac{w_i}{\sqrt{i}} + \frac{1}{2} \max_{i \leq t} w_i \iota + \sqrt{2\iota \sum_{i=1}^{t} w_i^2}$$

$\square$

The rest of this section is devoted to the proofs of the Lemmas used in the proofs of Lemma 17. We begin the following useful lemma adapted from Lemma 1 in [21], which is crucial in constructing high probability guarantees.

**Lemma 18.** *For any sequence of coefficients $c_1, c_2, \ldots, c_t$ s.t. $c_i \in [0, 2\gamma_i]^A$ is $\mathcal{F}_i$-measurable, we have with probability $1 - p/AK$,*

$$\sum_{i=1}^{t} w_i \left\langle c_i, \hat{l}_i - l_i \right\rangle \leq \max_{i \leq t} w_i \iota$$

*Proof.* Define $w = \max_{i \leq t} w_i$. By definition,

$$
\begin{aligned}
w_i \hat{l}_i(a) &= \frac{w_i \tilde{l}_i(a) \, \mathbb{I}\{a_i = a\}}{\theta_i(a) + \gamma_i} \leq \frac{w_i \tilde{l}_i(a) \, \mathbb{I}\{a_i = a\}}{\theta_i(a) + \frac{w_i \tilde{l}_i(a) \mathbb{I}\{a_i = a\}}{w} \gamma_i} \\
&= \frac{w}{2\gamma_i} \frac{\frac{2\gamma_i w_i \tilde{l}_i(a) \mathbb{I}\{a_i = a\}}{w \theta_i(a)}}{1 + \frac{\gamma_i w_i \tilde{l}_i(a) \mathbb{I}\{a_i = a\}}{w \theta_i(a)}} \overset{(i)}{\leq} \frac{w}{2\gamma_i} \log\left(1 + \frac{2\gamma_i w_i \tilde{l}_i(a) \, \mathbb{I}\{a_i = a\}}{w \theta_i(a)}\right)
\end{aligned}
$$

where $(i)$ is because $\frac{z}{1+z/2} \leq \log(1 + z)$ for all $z \geq 0$.

Defining the sum

$$\hat{S}_i = \frac{w_i}{w} \left\langle c_i, \hat{l}_i \right\rangle, \quad S_i = \frac{w_i}{w} \left\langle c_i, l_i \right\rangle,$$

we have

$$\mathbb{E}_i\left[\exp\left(\hat{S}_i\right)\right] \le \mathbb{E}_i\left[\exp\left(\sum_a \frac{c_i\left(a\right)}{2\gamma_i}\log\left(1+\frac{2\gamma_i w_i \tilde{l}_i\left(a\right)\mathbb{I}\left\{a_i=a\right\}}{w\theta_i\left(a\right)}\right)\right)\right]$$

$$\overset{(i)}{\le} \mathbb{E}_i\left[\prod_a\left(1+\frac{c_i\left(a\right)w_i\tilde{l}_i\left(a\right)\mathbb{I}\left\{a_i=a\right\}}{w\theta_i\left(a\right)}\right)\right]$$

$$= \mathbb{E}_i\left[1+\sum_a\frac{c_i\left(a\right)w_i\tilde{l}_i\left(a\right)\mathbb{I}\left\{a_i=a\right\}}{w\theta_i\left(a\right)}\right]$$

$$= 1+S_i \le \exp\left(S_i\right)$$

where $(i)$ is because $z_1\log\left(1+z_2\right) \le \log\left(1+z_1 z_2\right)$ for any $0 \le z_1 \ge 1$ and $z_2 \ge -1$. Here we are using the condition $c_i\left(a\right) \le 2\gamma_i$ to guarantee the condition is satisfied.

Equipped with the above bound, we can now prove the concentration result.

$$\mathbb{P}\left[\sum_{i=1}^t\left(\hat{S}_i-S_i\right)\ge \iota\right] = \mathbb{P}\left[\exp\left[\sum_{i=1}^t\left(\hat{S}_i-S_i\right)\right]\ge \frac{AK}{p}\right]$$

$$\le \frac{p}{AK}\mathbb{E}_t\left[\exp\left[\sum_{i=1}^t\left(\hat{S}_i-S_i\right)\right]\right]$$

$$\le \frac{p}{AK}\mathbb{E}_{t-1}\left[\exp\left[\sum_{i=1}^{t-1}\left(\hat{S}_i-S_i\right)\right]E_t\left[\exp\left(\hat{S}_t-S_t\right)\right]\right]$$

$$\le \frac{p}{AK}\mathbb{E}_{t-1}\left[\exp\left[\sum_{i=1}^{t-1}\left(\hat{S}_i-S_i\right)\right]\right]$$

$$\le \cdots \le \frac{p}{AK}$$

The claim is proved by taking the union bound. $\qquad\square$

Using Lemma 18, we can bound the $(A)(B)(C)$ separately as below.

**Lemma 19.** *If $\eta_i \le 2\gamma_i$ for all $i \le t$, with probability $1-p$, for any $t \in [K]$ and $\theta^* \in \Delta^A$,*

$$\sum_{i=1}^t w_i\left\langle\theta_i-\theta^*,\hat{l}_i\right\rangle \le \frac{w_t\log A}{\eta_t}+\frac{A}{2}\sum_{i=1}^t\eta_i w_i+\frac{1}{2}\max_{i\le t}w_i\iota$$

*Proof.* We use the standard analysis of FTRL with changing step size, see for example Exercise 28.13 in [17]. Notice the essential step size is $\eta_t/w_t$,

$$\sum_{i=1}^t w_i\left\langle\theta_i-\theta^*,\hat{l}_i\right\rangle \le \frac{w_t\log A}{\eta_t}+\frac{1}{2}\sum_{i=1}^t\eta_i w_i\left\langle\theta_i,\hat{l}_i^2\right\rangle$$

$$\le \frac{w_t\log A}{\eta_t}+\frac{1}{2}\sum_{i=1}^t\sum_{a\in\mathcal{A}}\eta_i w_i\hat{l}_i\left(a\right)$$

$$\overset{(i)}{\le} \frac{w_t\log A}{\eta_t}+\frac{1}{2}\sum_{i=1}^t\sum_{a\in\mathcal{A}}\eta_i w_i l_i\left(a\right)+\frac{1}{2}\max_{i\le t}w_i\iota$$

$$\le \frac{w_t\log A}{\eta_t}+\frac{A}{2}\sum_{i=1}^t\eta_i w_i+\frac{1}{2}\max_{i\le t}w_i\iota$$

where $(i)$ is by using Lemma 18 with $c_i(a)=\eta_i$. The any-time guarantee is justifed by taking union bound. $\qquad\square$

**Lemma 20.** *With probability* $1 - p$, *for any* $t \in [K]$,

$$\sum_{i=1}^{t} w_i \left\langle \theta_i, l_i - \hat{l}_i \right\rangle \leq A \sum_{i=1}^{t} \gamma_i w_i + \sqrt{2\iota \sum_{i=1}^{t} w_i^2}$$

*Proof.* We further decopose it into

$$\sum_{i=1}^{t} w_i \left\langle \theta_i, l_i - \hat{l}_i \right\rangle = \sum_{i=1}^{t} w_i \left\langle \theta_i, l_i - \mathbb{E}_i \hat{l}_i \right\rangle + \sum_{i=1}^{t} w_i \left\langle \theta_i, \mathbb{E}_i \hat{l}_i - \hat{l}_i \right\rangle$$

The first term is bounded by

$$\sum_{i=1}^{t} w_i \left\langle \theta_i, l_i - \mathbb{E}_i \hat{l}_i \right\rangle = \sum_{i=1}^{t} w_i \left\langle \theta_i, l_i - \frac{\theta_i}{\theta_i + \gamma_i} l_i \right\rangle$$

$$= \sum_{i=1}^{t} w_i \left\langle \theta_i, \frac{\gamma_i}{\theta_i + \gamma_i} l_i \right\rangle \leq A \sum_{i=1}^{t} \gamma_i w_i$$

To bound the second term, notice

$$\left\langle \theta_i, \hat{l}_i \right\rangle \leq \sum_{a \in \mathcal{A}} \theta_i(a) \frac{\mathbb{I}\{a_t = a\}}{\theta_i(a) + \gamma_i} \leq \sum_{a \in \mathcal{A}} \mathbb{I}\{a_i = a\} = 1,$$

thus $\{w_i \left\langle \theta_i, \mathbb{E}_i \hat{l}_i - \hat{l}_i \right\rangle\}_{i=1}^{t}$ is a bounded martingale difference sequence w.r.t. the filtration $\{\mathcal{F}_i\}_{i=1}^{t}$. By Azuma-Hoeffding,

$$\sum_{i=1}^{t} \left\langle \theta_i, \mathbb{E}_i \hat{l}_i - \hat{l}_i \right\rangle \leq \sqrt{2\iota \sum_{i=1}^{t} w_i^2}$$

$\square$

**Lemma 21.** *With probability* $1 - p$, *for any* $t \in [K]$ *and any* $\theta^* \in \Delta^A$, *if* $\gamma_i$ *is non-increasing in* $i$,

$$\sum_{i=1}^{t} w_i \left\langle \theta^*, \hat{l}_i - l_i \right\rangle \leq \max_{i \leq t} w_i \iota / \gamma_t$$

*Proof.* Define a basis $\{e_j\}_{j=1}^{A}$ of $\mathbb{R}^A$ by

$$e_j(a) = \begin{cases} 1 \text{ if } a = j \\ 0 \text{ otherwise} \end{cases}$$

Then for all the $j \in [A]$, apply Lemma 18 with $c_i = \gamma_t e_j$. Sine now $c_i(a) \leq \gamma_t \leq \gamma_i$, the condition in Lemma 18 is satisfied. As a result,

$$\sum_{i=1}^{t} w_i \left\langle e_j, \hat{l}_i - l_i \right\rangle \leq \max_{i \leq t} w_i \iota / \gamma_t$$

Since any $\theta^*$ is a convex combination of $\{e_j\}_{j=1}^{A}$, by taking the union bound over $j \in [A]$, we have

$$\sum_{i=1}^{t} w_i \left\langle \theta^*, \hat{l}_i - l_i \right\rangle \leq \max_{i \leq t} w_i \iota / \gamma_t$$

$\square$

## Footnotes

[4]In [37] the states are denoted by $\{i_a, i_b\}_{i=2}^H$ instead. Here we slightly change the notation to make it different from the notation of the actions