[Reviews · NeurIPS 2020]

Review 1

Summary and Contributions: The paper presents three theoretical results, focusing on sample-efficient learning in two-player zero-sum model-free games with discrete states and discrete actions: - Optimistic Nash Q-Learning, that maintains an upper and lower bound on the Q-Value of every state-actionA-actionB tuple. This algorithms is proven to have a sample complexity of O(H^5 SAB) - Optimistic Nash V-Learning, that learns an upper bound on the V-function of every state, and iterates on the actions of the A agent to compute a loss, used to train a policy. This algorithm has a complexity of O(H^6 S(A+B)). - A theoretical result that learning a policy against an opponent with no regret cannot be done with a polynomial complexity.

Strengths: The theoretical results are strong and well-explained. They seem novel, and largely improve compared to the current state of the art. The two algorithms proposed in the paper allow to choose which ones suits a problem the best, depending on the length of an episode and the number of actions that the A and B agents have. The Appendix contains many proofs and additional details, useful to understand the main results presented in the paper.

Weaknesses: A "weakness" of the paper is that it introduces three contributions at once, and therefore is very compact. While the paper is well-written and easy to read for the amount of information it contains, many parts of the paper lack intuition (like Equation 5 and the whole meaning of alpha, and the impact on how it is computed), and many references to the Appendix make reading the paper a bit difficult at time. This is really a paper that the reader must accept to read while not understanding everything, even though the Appendix answers many questions after the paper is read.

Correctness: The proposed algorithms are sound, and attentive reading (even though I'm not an expert in the field of the paper) allows to understand the proofs, and see that they have a high probability of being correct.

Clarity: The paper is well-written, and the gist of it can be understood by a non-expert. However, parts of the paper lack intuition, and the paper does not really try/manage to teach the reader things he/she may not know yet.

Relation to Prior Work: Related work is both nicely presented in a dedicated section, and discussed in the paper while relevant. The authors take care to compare two-player MDPs with single-agent MDPs when needed, and when a single-agent result does not apply to the two-players setting.

Reproducibility: Yes

Additional Feedback: The "broader impact" section at the end of the paper, and the general discussion of potential future work and applications of this theoretical paper, is generally a bit lacking. One or two paragraphs, discussing how the results presented in this paper would translate to real-world applications (examples of applications with discrete states), or to continuous-action settings (how would we do it? What would still hold and what wouldn't?) would have been interesting, even if only very high-level. Author response: the authors acknowledge that the paper is fully theoretical. Because they do not propose a discussion about possible applications of this paper (which I understand, as the paper is already quite dense), my remark here above remains.


Review 2

Summary and Contributions: The authors study the self play RL setting and analyze algorithms with provable guarantees (when the the control is centralized, i.e, a single algorithm controls the actions of both players). Furthermore, the authors also give some hardness results which suggest that decentralized learning is generally hard for this setting. ** I read the author's response and decided to keep my score. Thanks for the clarifications.

Strengths: *) The setting is interesting, and the fact the algorithm achieves performance nearly as the lower bound (up to factors of the horizon) is very impressive.

Weaknesses: *) Memory complexity. As far as I understood, the algorithms need to store K policies. This makes the algorithm impractical. I also suspect that in practice using only the last policies would improve the algorithm. *) The sampling of the policies. The authors suggest a policy certification algorithm from which they can read the final policies on which they prove the PAC guarantees. I felt that this procedure is not explained well enough (it is discussed in half a page, however, it seems to me there is not much of a discussion on why this procedure works). For example, there is no discussion on the question whether the policy certification procedure is necessary or why there is no 'natural' way to obtain a policy beside of performing this procedure. I believe that the lack of clarity on this subject harms the quality of the paper. *) What is the reason to include algorithm 1 in the paper? it seems it results in worse performance relatively to algorithm 2 and does not improve it in any other way.

Correctness: *) I've went over the proofs and they seem correct. I suggest the authors to add some references when using standard regret analysis techniques, or explain it in a bit more clarity (e.g., stating that \sum_i^N 1/\sqrt{i}\leq \sqrt{N}, and reference to the q-learning paper in 439 when changing the summation indices.) Think it will ease the reading for people that are less familiar with these techniques.

Clarity: The paper is clearly written.

Relation to Prior Work: Yes.

Reproducibility: Yes

Additional Feedback: *) Is there a reason to mention algorithm 1? it seems algorithm 2 gives improved performance relatively to it. If so, why presenting the two algorithms and not just algorithm 2? *) Although equation 9 can be thought of as a set of n+m linear constraints, why the optimization problem is always feasible? meaning, for any P and Q. *) I think the authors should elaborate more on the policy certification procedure. Although the authors devoted half a page to explain on this procedure, I feel it is not well explained. Most of the discussion is not devoted to explaining the policy certification procedure. Small comments. *) Line 151. Why for a fixed \mu the best response is not markovian? isn't it true that when mu is fixed minimizing over nu means solving an MDP (for which exists an optimal markovian policy)? *) Line 213. Where are \hat mu and \hat nu defined? I only see reference to \mu_k \nu_k. *) Line 239. Why using D_kl(\mu | \mu_0) instead of the entropy? I think the relation to FTRL is then a bit more apparent. *) Line 449. 'na'-> an? *) Line 455. 'hostory'-> history. *) Why the last equality relation in page 15 holds?


Review 3

Summary and Contributions: This work proves that for two player zero sum games, an optimistic variant of Nash Q learning can approximate nash equilibrium with sample complexity O(SAB), and an optimistic variant of Nash V learning with sample complexity O(S(A+B)), improved from previous results O(S^2AB). Additionally, achieving sublinear regret when playing against adversarial opponents in Markov games is proved to be hard and non-polynomial.

Strengths: This is the first time that sample complexity in two player zero-sum game to reduce to O(S(A+B)), which matches theoretical lower bound expect polynomial of episode length. The algorithm itself is described clearly, which is built upon Coarse Correlated Equilibrium subroutine and Follow-the-Regularized-Leader algorithm. A novel approach to compute the certified policies given the near-optimal Nash equilibrium values are also provided, for both Nash Q learning and Nash V learning. The new bound is a significant contribution in terms of sample complexity.

Weaknesses: A major drawback is that not a single empirical experiment is done, in contract to a few prior works. e.g. Nash Q-Learning for General-Sum Stochastic Games, J. Hu and M. P. Wellman. At least experiments in matrix games or grid world should be conducted, to verify the correctness of the proof. It will be great to see that sample complexity increase linearly with S, A, B with detailed ablation studies, and what percentage of runs reach eps-nash with the new algorithms. It is also not clear that why the algorithm can bring the sample complexity down. What is the intuition that the bound can be reduced? What is the intuition to pick the a_t and b_t and other hyperparameters in the formulas? Since the algorithm is built upon Q learning, how can this algorithm extend to other more general cases? Is there any intuitions that can help with real world algorithms in terms of sample complexity? === Post rebuttal: Some of the intuitions are explained. After the discussion with other reviewers I still think this paper needs to be improved with presentation and preliminary experiments if possible.

Correctness: See weaknesses.

Clarity: The presentation of this paper is slightly unclear, many details of in the proofs and not explained clearly about the intuitions in the main text. Also Sec. 5 seems to be slightly disconnected to the rest of the paper.

Relation to Prior Work: Yes, it compares the new algorithm with prior related works in terms of sample complexity of S, A, B. The authors build upon existing algorithms CCE and FTRL, and with the proposed new algorithm the sample complexity can be reduced.

Reproducibility: Yes

Additional Feedback:

[Author Response · NeurIPS 2020]

We thank all reviewers for their valuable feedback. We would like to first reiterate our main contribution of this paper,
and then respond to the individual reviews.

**Main Contribution:** The main focus of this paper is to gain fundamental understanding of reinforcement learning
in the setting of two-player zero-sum games, and especially to investigate the fundamental question "*what is the*
*minimum amount of samples required for provable learning?*". Our results for the first time close this important open
problem—i.e. the optimal sample complexity of learning Markov games—in all parameters except episode length. The
sample complexities of our newly proposed algorithms also dramatically improve upon the existing ones (see Table 1).
We believe our results make a significant contribution to the field of theoretical reinforcement learning.

**Reviewer #1**. We thank reviewer 1 for the overall positive feedback. Due to the space limit of NeurIPS, we have to cut
some explanations/intuitions and defer some to appendix. We will try to provide better explanations and rearrange the
materials in the final version.

**Reviewer #2**. 1. *Memory complexity.* The main focus of this paper is the sample efficiency to learning a Markov game
(see the second paragraph of rebuttal). As a tradeoff for obtaining the near-optimal sample complexity, we agree our
memory complexity is not completely intractable but still undesirable at the current stage. There are several possible
algorithmic ideas that we believe may help improve this memory complexity (for instance, the low-switching idea in
"Provably Efficient Q-Learning with Low Switching Cost" paper). Given the amount of results already contained in this
paper, we leave memory efficiency for future work.

2. *Why there is no 'natural' way to obtain a policy beside of performing this procedure.* The most standard way we
are aware of in the literature of MDP is to directly use the policies the algorithm used in the last episode (or a random
episode) during the training process. In online algorithms such as our Nash Q-learning, those policies are guaranteed to
perform well only against the Nash equilibrium, and not the best response (which is what we desire). Certified policy is
one way we design to fix this problem, but may not be the only way or "necessary". Whether there exists any alternative
way remains open, and we believe it is an interesting direction to explore in the future.

3. *Why include algorithm 1.* We include algorithm 1—Optimistic Nash Q-learning—because (1) Nash Q-learning
(without optimism) itself is a well-known classical algorithm whose non-asymptotic theoretical guarantee remains
absent. Therefore, analyzing a variant of Nash Q-learning may be of independent interest. (2) Algorithm 3 is built upon
several algorithmic ideas behind Algorithm 1. They share many common traits, like incremental (model-free) update of
value functions, and certified policy. Therefore, algorithm 1 serves as a warmup version to hopefully help the reader
understand Algorithm 3, which is more sample efficient but also more involved.

4. *Why optimization problem of equation 9 is always feasible.* Equation 9 is not an optimization problem, and we
assume the reviewer actually means equation 8. As mentioned in line 397-399, Nash equilibrium (NE) is a special case
of CCE. Since NE always exists, CCE always exists, i.e., the set of linear constraints are always feasible.

5. *Why best response of fixed $\mu$ can be non-Markovian?* Only when $\mu$ is fixed and *Markovian*, minimizing over $\nu$ means
solving an MDP. However, when $\mu$ is *non-Markovian*, the best response $\nu$ can be non-Markovian and dependent on
the history. For example, if the max-player follow the strategy that chooses a random action in the first step, and then
always pick the same action (same as the first one) in all later steps, the best response would not only depends on the
current state, but also depends on the action of the max-player in the first step, thus non-Markovian.

6. *Definition of $\hat{\mu}$ and $\hat{\nu}$.* This is defined in Algorithm 2 for Nash Q-learning and in Algorithm 4 for Nash V-learning.
The "hat" version is the actual certified policy (which can be executed as in Algorithm 2 and 4).

**Reviewer #3**. 1. *A major drawback is that not a single empirical experiment is done.* The main focus of this paper is
theoretical, and we believe our theoretical contribution is significant (see the second paragraph of rebuttal). On one
hand, we agree experiment/practical performance is important, and it is definitely worth further investigation. On the
other hand, given the large amount of pure theoretical ML work without experiments published every year at ICML and
NeurIPS, we hope our paper can be evaluated based on its theoretical significance.

2. *Intuitions behind the improvement.* The short answer for the reason behind the improvement is that: Nash Q-learning
is an online/incremental update algorithm, which avoids the complicated statistical dependency among the data as in
previous algorithm [2], thus shaving off an $S$ factor. This is also briefly explained in line 219-223. Nash V-learning
deploys the idea of follow-the-regularized-leader per step, which provides regret guarantee regardless of the number of
actions of the opponent, thus reducing the sample complexity from $AB$ to $A + B$. This is also briefly explained in line
239-243. We will add more explanations in the future version.

[Meta-Review · NeurIPS 2020]

After reading the reviews and authors' responses, it seems the only main concern raised is the lack of experiments. My opinion is that while experiments would be nice to have, the lack of experiments is not a significant concern if the theoretical results are strong enough. In my own assessment of the paper, I find the theoretical results to be indeed quite a strong contribution to the field (they provide the first algorithm to match the PAC lower bound, for a problem which has quite a few previous works). The reviewers seem to agree with this point in their reviews. I, therefore, recommend that the paper be accepted.